# Meta-programmable analog differentiator

Jérôme Sol[1], David R. Smith[2] & Philipp del Hougne [3✉]

We present wave-based signal differentiation with unprecedented fidelity and flexibility by purposefully perturbing overmoded random scattering systems such that zeros of their scattering matrices lie exactly at the desired locations on the real frequency axis. Our technique overcomes limitations of hitherto existing approaches based on few-mode systems, both regarding their extreme vulnerability to fabrication inaccuracies or environmental perturbations and their inability to maintain high fidelity under in-situ adaptability. We demonstrate our technique experimentally by placing a programmable metasurface with hundreds of degrees of freedom inside a 3D disordered metallic box. Regarding the integrability of wave processors, such repurposing of existing enclosures is an enticing alternative to fabricating miniaturized devices. Our over-the-air differentiator can process in parallel multiple signals on distinct carriers and maintains high fidelity when reprogrammed to different carriers. We also perform programmable higher-order differentiation. Conceivable applications include segmentation or compression of communication or radar signals and machine vision.

---

[1] INSA Rennes, CNRS, IETR - UMR 6164, F-35000 Rennes, France. [2] Center for Metamaterials and Integrated Plasmonics, Department of Electrical and Computer Engineering, Duke University, Durham, NC 27708, USA. [3] Univ Rennes, CNRS, IETR - UMR 6164, F-35000 Rennes, France. ✉email: philipp.del-hougne@univ-rennes1.fr

Differentiation is a pivotal mathematical operation with signal processing applications including edge-based segmentation for data compression or image sharpening, FM-to-AM demodulation for communication or Doppler-radar processing, as well as machine vision and hearing. Analog implementations of differentiation on wave processors promise much higher speeds and minimal power consumption in contrast to their digital electronic counterparts[1,2]. Yet, current wave-based differentiators are plagued by excessive sensitivity to fabrication inaccuracies and environmental conditions, and lack in-situ adaptability. In this paper, instead of fabricating a carefully designed single-mode structure, we take an overmoded random scattering system as starting point and show that purposeful perturbations of its scattering properties, here with hundreds of degrees of freedom offered by an array of programmable meta-atoms[3,4], enable unprecedented flexibility and fidelity of over-the-air wave-based differentiators by imposing, at will and in situ, that zeros of the scattering matrix lie exactly on the real frequency axis.

Any linear wave system's input-output relation is a linear transformation that can be either a discretized or a continuous operator. The input and output channels can originate from various degrees of freedom, such as space (guided or propagating in free space), frequency and polarization, or even a mix thereof. The needs of artificial intelligence have recently driven the implementation of arbitrary matrix multiplications[5–11], mostly for discretized spatial channels[12]. At the same time, basic mathematical operations like differentiation continue to be of fundamental importance, for instance, in machine and human vision[13] for data compression via edge detection. These mathematical operations are typically implemented as continuous operators. A differentiator is a linear filter whose transfer function, $H(\omega) = i(\omega - \omega_0)$ for first-order differentiation of the envelope of a carrier at $\omega_0$, can be found in many natural and artificial special physical near-zero scattering scenarios. If information is encoded spatially such as in an optical image, then critical-coupling (CC) settings[14,15], the Brewster effect[16], and various (metamaterial) layer structures[17–20] have been shown to yield this transfer function. For temporally encoded information, implementations based on fiber gratings[21–24], critically coupled microring resonators[25], directional couplers[26], or interferometers[27–29] have been put forward. All these settings require very careful fabrication and/or alignment and are hence highly vulnerable to inaccuracies in the fabrication or operation environment. Wave-based differentiators are some of the most vulnerable wave processors because the underlying wave system is operated at a scattering anomaly ($H(\omega_0) = 0$); the origin of this extreme sensitivity can be traced back to a fundamental quantity in mesoscopic physics: the dwell time of the wave in the system[30]. We will elaborate on this link for the concrete case of CC for temporal differentiation in the following, which is also directly relevant to our experiments.

A longstanding question in wave physics and material science is how to excite a structure such that no wave energy is reflected back. CC is the simplest case of zero reflection and usually refers to coupling a single (guided) incident channel to an isolated mode of the structure[31,32], requiring perfect matching of the structure's excitation and decay rate. Moreover, perfect absorption has also been demonstrated for normal incidence of plane waves on thin metamaterials[33–35]. A generalized version of CC is coherent perfect absorption[36–40] (CPA) of multi-channel radiation by a resonant structure (with possibly overlapping modes). In all these cases, one zero of the scattering matrix is real-valued such that the structure can act as a steady-state sink: all incident radiation can be perfectly absorbed (see also Supplementary Note 1A). Since this zero's imaginary part vanishes, the dwell time of the corresponding wavefront in the system diverges[36,41–43] (see also Supplementary Note 1B). The latter makes for extremely sensitive detectors[42,44], but conversely means that minute detuning of any system parameter moves the zero away from the real axis. As soon as the zero leaves the real axis, the spectrum's linear V shape and the associated abrupt $\pi$ phase jump in the vicinity of $\omega_0$ are no longer possible. Consequently, minute imperfections severely deteriorate and rapidly undermine the faithfulness of an analog differentiator, to the point that it becomes unsatisfactory, as illustrated in detail in Supplementary Note 3. This clarifies why tiny fabrication or alignment imperfections, or minute environmental perturbations, can severely impact the fidelity of a wave-based differentiator. Similarly, adapting, for instance, the filter's $\omega_0$ in situ in order to consecutively process envelopes on different carriers is generally very difficult or impossible. Wave processors realizing other functionalities which do not involve diverging dwell times are expected to display significantly less sensitivity.

To overcome the excessive sensitivity of wave processors, a recent trend has been to explore ideas from topological photonics[45–47] but the lack of in-situ adaptability remains. In parallel, the switching between different functionalities (differentiation, equation solving, etc.) has been studied[29,48,49] but this does not offer in-situ adaptability of a given functionality, e.g., to the signal carrier, nor robustness to fabrication inaccuracies or environmental perturbations. A clear route to address the aforementioned challenges are wave processors that can be reprogrammed in situ. A notable application thereof to temporal differentiation involved a photonic integrated interferometer equipped with programmable phase modulators[29]. On paper, it appears that a single degree of freedom is sufficient to introduce a relative phase shift of $(2m + 1)\pi$ at $\omega_0$ between the two interferometer arms[29], yielding a perfect differentiator. In practice, however, the transfer function's magnitude minimum in Ref. [29] appears to be around 0.16; this system has a zero close to rather than on the real frequency axis, which jeopardizes a linear V shape and the associated abrupt $\pi$ phase jump. Such limitations are inherent in the use of a single or few degrees of freedom in systems with a single or few resonances, and these limitations are also found in tunable microwave notch filters[50–53] (see also Supplementary Note 1C). The limitations originate from unaccounted non-idealities in real-life implementations such as coupling dispersion, transmission-line length variation, parasitic coupling, and other properties of the microstrip lines. Using a single or few degrees of freedom, it is certainly possible to tune few-resonance systems such that the zeros move in the complex plane; however, only under special conditions of $\mathcal{PT}$-symmetry[54,55] that are certainly not met by simple tunable notch filters (the uncompensated presence of absorption already trivially breaks $\mathcal{PT}$-symmetry) there is a guarantee that the zeros move exactly on the real frequency axis upon tuning. Hence, upon tuning simple notch filters in practice, zeros do not remain on the real frequency axis, and a few degrees of freedom tend to be insufficient to prevent that they drift away from the real frequency axis.

In contrast, our approach offers at least two orders of magnitude more degrees of freedom. This massive increase in programmability, together with the high density of zeros inherent to overmoded random scattering systems, makes it easy for us to perturb the system such that one of its zeros is placed, at a desired frequency, exactly on the real frequency axis with extremely high precision (notch depth $< -70\,\mathrm{dB}$). Moreover, we can also switch to different functionalities and simultaneously create multiple zeros at arbitrary frequencies (within the programmable metasurface's operating band) which is of importance for parallel wave processing at distinct frequencies, exploiting the wave equation's linearity[56,57]. Unlike conventional electronic processors, a single wave processor can simultaneously process various streams of information encoded on independent (spectral, polarization, etc.)

channels[49,57–59], directly multiplying its effective speed by the number of independent channels.

Our approach to rely on potentially bulky complex scattering systems such as a 3D disordered metallic box for microwave carriers may, at first sight, appear to be at odds with substantial efforts from the metamaterials community to miniaturize wave processors[17,60–63]. Indeed, the bulkiness of early optical processors, which relied on free-space propagation[64,65], but also the need for additional refractive elements (prisms, lenses) of many recent flat-optics designs, thwarts integrability. Our technique, however, does not have any special requirements regarding the complex scattering system such that it can be implemented based on any already existing bulky system whose primary functionality is not related to signal processing. We envision that ubiquitous metallic enclosures, such as a military toolbox or a microwave oven, can be endowed with a second signal processing functionality simply by inserting an ultrathin programmable metasurface[3,4] at an arbitrary location in order to tune their scattering properties. Thereby, we introduce a new perspective on integrability that decouples it from device volume and related miniaturization efforts. As such, our approach is arguably at least as convenient regarding integrability as are miniaturized wave processors. Its strength becomes particularly apparent for processing long-wavelength signals such as microwaves or sound in their native analog domain. In such scenarios relevant to radar, wireless communication, gesture recognition, ambient-assisted living, untethered virtual reality or voice-commanded devices, even metamaterial-based processors are cumbersome whereas our technique only requires the user to place an ultrathin programmable metasurface[3,4] at an arbitrary location inside an existing enclosure. A related alternative view on the integrability of wave processors was introduced in Ref. [9], but for less vulnerable spatially discrete monochromatic arbitrary matrix multiplications. Moreover, in Ref. [9] the configuration of the programmable metasurface was interpreted as input, that is, the wave processor did not operate in the signal's native domain, and furthermore averaging over multiple realizations was necessary as a consequence. If specific applications require far more compact implementations, it is also possible to implement our technique in flat quasi-2D programmable chaotic cavities with currently available technology[66].

The technique underlying the present work to purposefully perturb a complex scattering system to impose real-valued scattering zeros on demand is, on the one hand, conceptually a topic of significant contemporary interest in mesoscopic physics. The concept was proposed and experimentally demonstrated in Ref. [44] for single-channel CPA in a lossy chaotic microwave cavity, with applications to precision sensing and secure communication. The idea was subsequently generalized to multiple channels[42,67], including cases with additional constraints on the allowed input wavefront[68]. The notion of parameter tuning is also central to various other recent works on scattering anomalies[55,69]. On the other hand, our specific experimental implementation in the microwave domain leveraging programmable metasurfaces relates to a large body of literature: these arrays of meta-atoms with individually reconfigurable scattering properties (usually reflection coefficient) are primarily used for free-space applications such as adaptive beamforming[3], holography[70], diffuse scattering[71], wireless communication[72–74], (intelligent) imaging[66,75–80] and spatio-temporal wave control[81]. However, they also find increasingly use inside rich-scattering environments[82] as evidenced by various experiments on focusing[83–86], (sub-wavelength) sensing[87–89] and transmission matrix engineering[9,90,91]. Nonetheless, the generality of the discussed wave concepts implies that our idea can also be implemented in tunable acoustic or optical scattering systems[92–95].

In this paper, we establish the unprecedented flexibility offered for wave-based differentiators based on judiciously tuned over-moded random scattering systems, through a series of microwave experiments in a prototypical metallic disordered box equipped with 1-bit programmable metasurfaces. Our work is realized in the WLAN 5-GHz band and hence of potentially direct technological relevance. First, we demonstrate our ability to impose in situ a scattering zero at any desired frequency in this band, and to toggle at will between them. We directly inject various complicated waveforms in time-domain experiments to provide direct experimental evidence of our ability to compute the temporal derivative of an envelope of an arbitrary carrier (within the 5-GHz band). Second, we generalize this concept to toggling between multiple simultaneous zeros of the scattering matrix— that lie on the real frequency axis—of various spectral separations in order to demonstrate parallel wave processing on the same device. Finally, we cascade two tunable chaotic cavities to compute second-order derivatives, thereby engineering a transfer function usually associated with CPA exceptional points[96–99] that may also be enticing for applications in broadband near-perfect absorption.

## Results

**Operation principle.** To start, our goal is to compute the temporal derivative of a function $e(t)$ that modulates an arbitrary carrier $\omega_0$ by letting it bounce off the interface between a port and an irregularly shaped electrically large metallic box—see Fig. 1. For the differentiation operation to happen, the system's transfer function $H(\omega)$ (i.e., the port's reflection spectrum) should approximate $i(\omega - \omega_0)$ in the vicinity of $\omega_0$, as sketched in the inset. A zero of the scattering matrix that lies on the real frequency axis at $\omega_0$, which we define as a real-valued zero, yields this functional form of the reflection spectrum upon excitation[42,97]. However, the chances of observing a real-valued zero at the desired frequency $\omega_0$ within a random scattering system are extremely low[44]. By tuning a continuous system parameter, it is possible to observe a real-valued zero but most likely not at the targeted operating frequency[55]. Yet, our goal is to be able to re-program our analog wave-based differentiator in situ depending on the use case; specifically, as shown in Fig. 1, we would like to adapt its transfer function to the incident carrier. To that end, we must be able to perfectly place a zero on the real-frequency axis on demand at any desired frequency.

To accomplish this goal, we mount two programmable metasurfaces[3,4] on the walls of the disordered box at arbitrary locations. These metasurfaces are ultrathin arrays of meta-atoms whose scattering properties can be controlled via a simple bias voltage. Our generic technique is not limited to implementations based on a specific programmable metasurface design, all that matters is that the programmable meta-atoms control as many rays as possible. Each meta-atom of the utilized prototype (see Supplementary Note 4 for technical details) has two possible states and is capable of roughly mimicking Dirichlet or Neuman boundary conditions under normal incidence[84]. By judiciously choosing the coding matrix of the programmable metasurface (see Supplementary Note 5), it is possible to impose a real-valued zero on demand at a desired frequency[44]. For instance, if the incident carrier is $\omega_1$ (blue), metasurface configuration $C_1$ (blue) is used; if the incident carrier is $\omega_2$ (red), metasurface configuration $C_2$ (red) is used, etc.

**Direct observation of meta-programmable differentiation.** We begin by identifying eight metasurface configurations that yield a real-valued scattering matrix zero at eight distinct, evenly spaced frequencies between 5.05 GHz and 5.4 GHz (see Methods). The

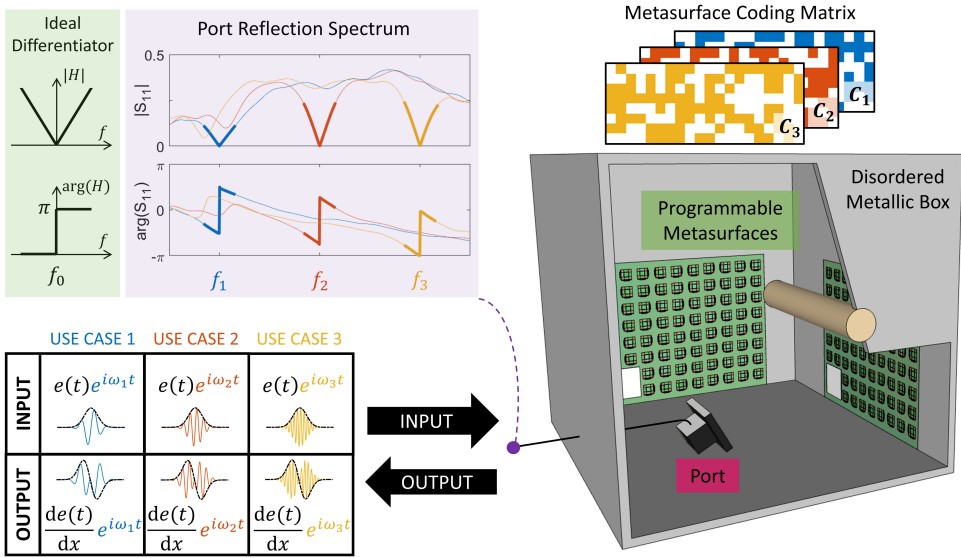

**Fig. 1 Operation principle of the meta-programmable analog temporal differentiator.** A signal $e(t)$ is modulated as envelope onto a carrier $\omega_0$ and incident via a guided single-mode channel on a disordered metallic electrically large box. The scattering properties of the latter can be tuned via programmable metasurfaces mounted on its walls such that the port's reflection spectrum has a zero at the carrier frequency. Then, the port's transfer function matches that of an ideal differentiator, displaying in the zero's vicinity a linear V shape centered on $\omega_0$ as well as an abrupt $\pi$ phase jump at $\omega_0$ (see inset). Consequently, the reflected signal's envelope is the temporal derivative of $e(t)$. Incident and reflected signals are separated via a circulator, see Supplementary Note 5 for details. A computer program digitally controls the realized differentiation operation by toggling between different metasurface configurations (color-coded) depending on the current needs in terms of the incident signal's carrier frequency.

resulting transfer functions are shown in Fig. 2a, b. The desired linear V shapes, accompanied by abrupt phase shifts of $\pi$, are observed exactly at the eight chosen frequencies. In each case, the magnitude of the transfer function is below $-70$dB at the central frequency (see Supplementary Note 6). The bandwidth over which the transfer functions are a good approximation to an ideal differentiator is around 15 MHz (see Supplementary Note 7). A global phase drift is observed and distinct for each operating frequency—however, as explained in Supplementary Note 2, this does not impact the desired differentiator functionality. Relative to the operating frequency, our fractional bandwidth of $3 \times 10^{-3}$ outperforms a number of previously reported temporal differentiators; only Ref. [22], a static non-programmable differentiator, achieves a considerably larger fractional bandwidth (see Supplementary Note 7). In comparison to the previously reported reconfigurable temporal differentiator from Ref. [29], our device's fractional bandwidth is an order of magnitude larger. More importantly, our device implements true real-valued zeros as evidenced by the depth of the reflection dips, their magnitude's linearity in the vicinity of the central frequency and the abruptness of the phase jump.

A direct experimental observation of meta-programmable differentiation requires the injection of various waveforms modulated onto various carriers and the observation of their reflections. We modulate each of the considered carriers in turn with one of the three envelope functions shown in Fig. 2c, f, i. For a given carrier, we toggle the metasurface to its corresponding configuration, we let the signal impinge on the port-cavity interface, and we measure the reflected signal—technical details are provided in Supplementary Notes 4 and 5. The first waveform, a Gaussian pulse (0.1 μs duration, 10 MHz bandwidth), is a typical function to test the quality of analog differentiators. Its derivative, if correctly computed, should be perfectly symmetric and have exactly zero amplitude at its center (see Fig. 2d). In Fig. 2e we superpose the envelopes of the output signals measured in the eight different carrier use cases. It is apparent that they are all highly similar and extremely close to the

ideal analytical derivative. We also test our meta-programmable differentiator's performance with two more complicated functions: a quadratic polynomial and the skyline of Rennes, France. For these two functions, some less dominant spectral components lie outside the differentiator's operating bandwidth. In the former case (Fig. 2h), the measured output signals faithfully reproduce the linear slopes expected for the derivative of a quadratic function, as well as the expected amplitudes. In the latter case (Fig. 2k), the agreement with the analytical result is also very good; only for some very sharp peaks the measured magnitude is slightly below the expected one. Besides demonstrating differentiation based on a random scattering system, the major technological relevance of our work is the unprecedented fidelity and flexibility of our differentiator. In this section, we leveraged its flexibility to toggle between differentiation for different carriers; in the following sections, we will further leverage this flexibility for more advanced and unique features.

**Parallelized meta-programmable differentiation.** A unique but to date largely underexploited advantage of wave processors over their electronic counterparts is that a single device can simultaneously process multiple data streams thanks to the linearity of the wave equation[49,56–59]. Thereby, its computational efficiency in terms of equivalent digital operations per unit time can be multiplied by the number of independent data streams. The differentiation operation that we are concerned with in this paper can be parallelized, for instance, by encoding different data streams onto different carriers. The reflection spectrum of the system should then simultaneously present multiple real-valued scattering zeros corresponding to the targeted carrier frequencies. The experimental setup remains that from Fig. 1, except that the impinging wave is now the sum of multiple independent carriers, each modulated by an independent data stream. This principle is illustrated in Fig. 3a. Our parallelized meta-programmable analog differentiator can toggle between different simultaneously targeted carrier frequencies. As shown in Fig. 3a, for a first use case, a blue metasurface configuration may be used that imposes

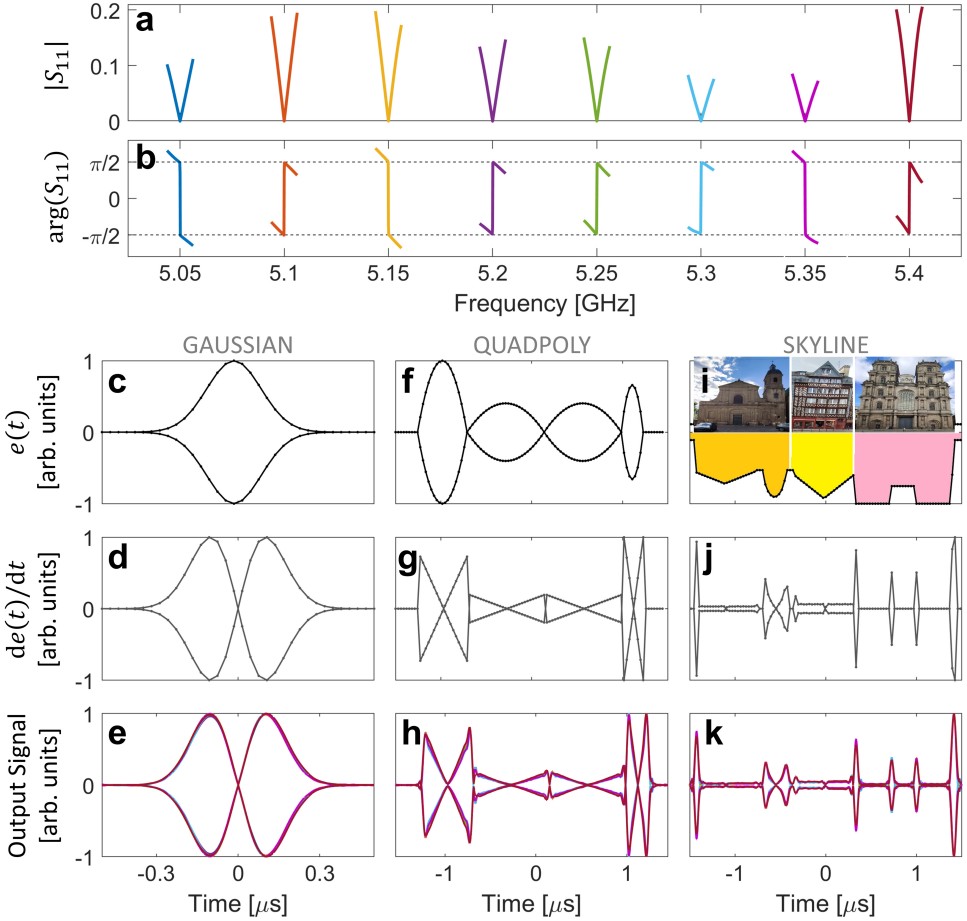

**Fig. 2 Direct experimental results of meta-programmable analog temporal differentiation. a**, **b** Amplitude (**a**) and phase (**b**) of the system's transfer function for eight different metasurface configurations (color-coded) that correspond to reflection zeros at eight equally spaced carrier frequencies in the 5-GHz band. **c**, **f**, **i** Envelopes $e(t)$ of experimentally injected waveforms corresponding to a Gaussian function (**c**), a set of quadratic polynomial functions (**f**), and a skyline of Rennes, France, composed of Basilique Saint-Sauveur, a traditional timber-framed house and Cathédrale Saint-Pierre (**i**). **d**, **g**, **j** Corresponding analytical derivatives $\frac{de(t)}{dt}$. **e**, **h**, **k** Corresponding experimentally measured output signals. For a given waveform and carrier, the metasurface is toggled to the configuration optimized for this carrier, the input signal is injected, and the reflected signal is measured. The envelopes of the measured output signals for all eight considered carriers are superposed on these figures using the same colors to identify different carriers as in (**a**, **b**).

simultaneous real-valued scattering zeros for carriers $\omega_{1,A}$ and $\omega_{1,B}$; for a second use case, a red metasurface configuration may be used for carriers $\omega_{2,A}$ and $\omega_{2,B}$, etc.

In Fig. 3b, c, f, g, j, k, three examples of experimental measurements of optimized transfer functions with two simultaneous real-valued zeros at different frequencies are shown. In all cases, the transfer function magnitude displays the linear V shape and the phase the corresponding abrupt $\pi$ jump. Examples with more than two simultaneous zeros are shown in Supplementary Note 9 but in the following we focus on two simultaneous zeros to directly test the meta-programmable parallelized differentiation ability of our system. To that end, we generate two independent signals and inject their sum into our system which is toggled to a state that corresponds to the two chosen carriers. We measure the reflected signal and bandpass-filter it around each of the two carriers, respectively. The resulting two demultiplexed data streams are shown in Fig. 3d, e, h, i, l, m. The obtained signal output envelopes match very well the analytically expected ones for both carriers in all three cases. These results illustrate our ability to perform parallelized meta-programmable differentiation. Ref. [29] based on a tunable interferometer did not discuss parallel computing but it is clear that if at all possible, the simultaneous real-valued zeros would have to be spaced according to the free spectral range as opposed to being able to

place them at arbitrary locations as in our system. If the carrier frequencies' separation exceeds our metasurface's operation bandwidth, separate metasurfaces (one for each carrier) may be deployed. In that case, only the designated metasurface would modulate the corresponding carrier frequency such that the different metasurface configurations could be optimized independently from each other[9].

**Higher-order meta-programmable differentiation**. To this point, we have considered only first-order temporal differentiation; in this section, we generalize the previous concepts to higher-order differentiation. An $n$-th-order differentiator's transfer function is $\left[i\left(\omega - \omega_0\right)\right]^n$, which can be physically implemented in various manners. For instance, instead of tuning our system to CPA, one could attempt to tune it to a CPA exceptional point (EP) at which two eigenvalues and eigenvectors of the wave operator associated with purely incoming boundary conditions coalesce[97]; such a CPA-EP would yield the transfer function associated with a second-order differentiator (see inset in Fig. 4). However, preliminary tests following this route were unsuccessful, indicating that more degrees of freedom and/or more elaborate optimization protocols may be necessary. If the input signal was known to have a sufficient temporal sparsity, a combination of directional couplers and delay loop could be

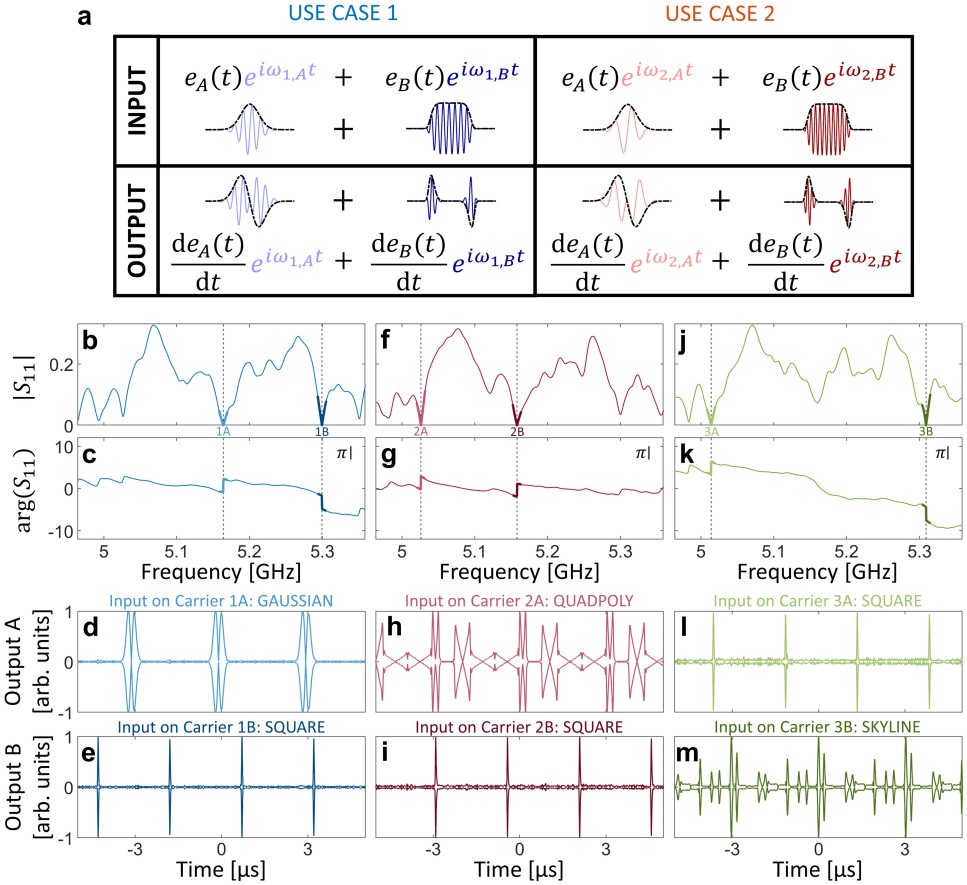

**Fig. 3 Parallelization of meta-programmable wave-based differentiation. a** Principle of parallel computing with spectral degrees of freedom. The injected waveform is the sum of two signals (two independent envelopes $e_A(t)$ and $e_B(t)$ modulated onto distinct carriers $\omega_{1,A}$ and $\omega_{1,B}$) and the reflected signal is the sum of the derivatives of these two envelopes, modulated onto the respective carriers. **b, c, f, g, j, k** Amplitude and phase of the system's transfer function for three examples (color-coded) of choices of two simultaneously imposed reflection zeros. The computer program can toggle between these by changing the metasurface configuration. **d, e, h, i, l, m** Envelope of experimentally measured output signal (spectrally bandpass-filtered around the respective carrier frequencies) upon injecting the indicated waveforms (see Supplementary Note 5 for details) in the three considered use cases.

added to the setup in Fig. 1 to realize the higher-order differentiation via multiple passes across the same port-cavity interface. Another route to obtain the desired transfer function of a second-order differentiator is to cascade two first-order differentiators. This more generic implementation corresponds to physically cascading multiple setups akin to the one from Fig. 1, all tuned to simultaneously have a real-valued scattering zero at the same frequency, as shown in Fig. 4a. We adopt the latter approach in the following.

Our goal is now to simultaneously impose a real-valued scattering zero at the same frequency in two similar but distinct chaotic cavities, and to toggle between different examples thereof that differ in terms of the chosen frequency, in order to then directly illustrate our ability to perform meta-programmable second-order differentiation. Given that our experimentally available number of programmable meta-atoms is limited, we now split them equally between the two cavities. The optimization problem is now much harder since we have only half of the previously available degrees of freedom in each cavity. This is of course not a general limitation of our concept but related to our experimental constraints, and we cater to it by allowing some flexibility regarding the chosen frequency in our optimization protocol. In the left columns of Fig. 4b–e, we report four examples of magnitude and phase of experimentally measured transfer functions for second-order differentiation. The flat phase (modulo $2\pi$ and ignoring the irrelevant background phase drift)

matches that of the ideal transfer function shown in the inset of Fig. 4a. The magnitudes also display the desired quadratic behavior in the vicinity of $\omega_0$ but appear to offer a more limited useable bandwidth than those from Fig. 2a. We directly test our meta-programmable second-order differentiator by injecting the Gaussian signal from Fig. 2c, modulated onto different carriers, and in each case toggling the metasurfaces to the configuration corresponding to the carrier. Compared to first-order differentiation, the output signals are much weaker and hence noisier, such that we average the output measurements over 20 acquisitions. The averaged output envelopes for the four considered carriers in Fig. 4b–e are close to the analytically expected output (see Fig. 4a). The symmetry with respect to the central peak is very good, and the two dips on either side go as low as possible given the measurement noise. The measurement noise is also evident far away from the central peak where the output signal is analytically zero.

Beyond the wave computing functionality discussed so far, the setup from Fig. 4 emulates, as previously noted, the transfer function of CPA-EPs which are thus far mainly investigated for broadband near-perfect absorption applications. In contrast to the intensity-dependent nonlinear CPA-EP in Ref. [99], our experiment offers this transfer function irrespective of the incident power signal. Moreover, the homogeneous absorption in our system due to Ohmic losses on the walls, as opposed to localized absorption mechanisms, enables the absorption of high-

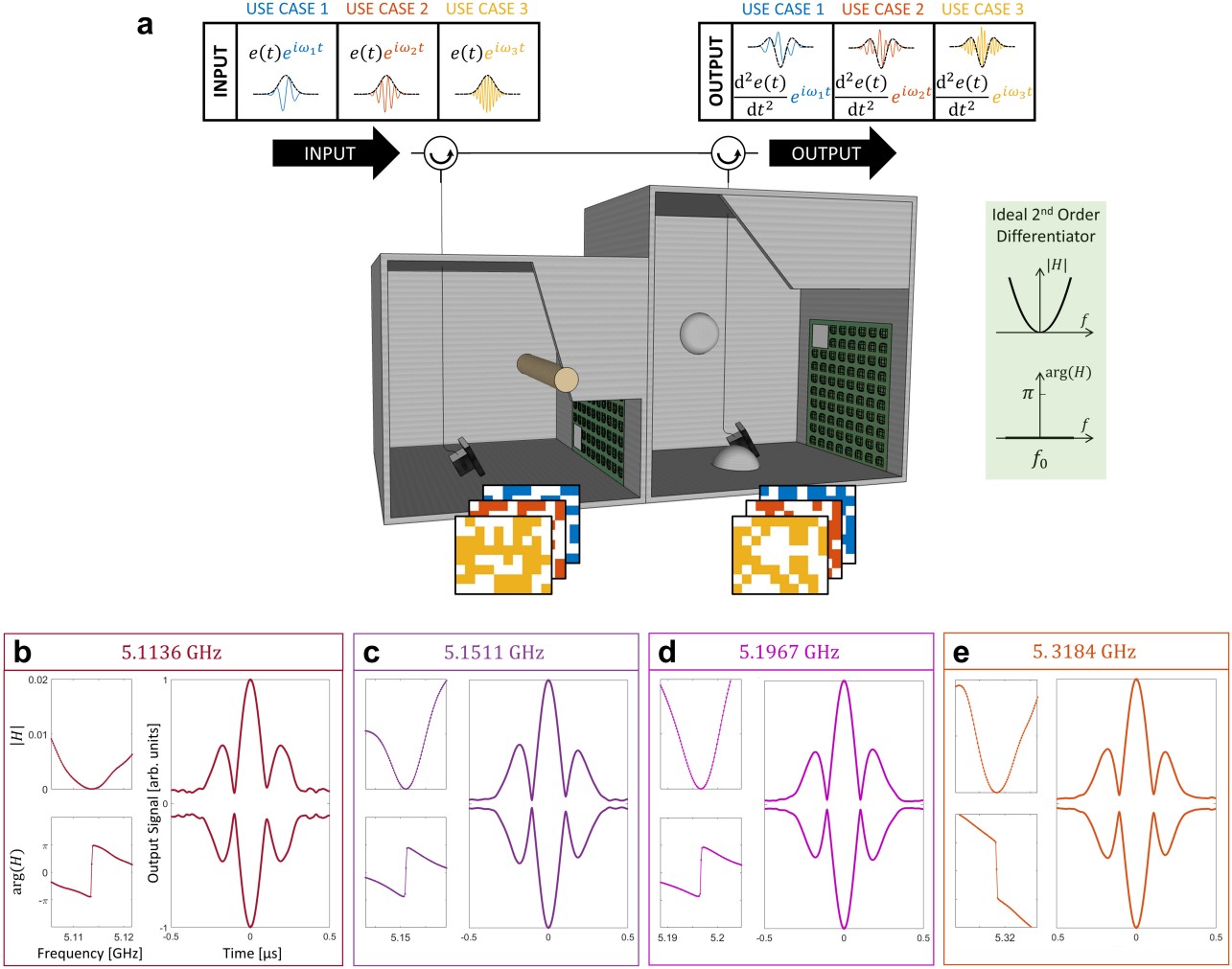

**Fig. 4 Meta-programmable second-order differentiator. a** Operation principle. Two setups akin to the one from Fig. 1 are cascaded (using circulators as shown) in order to implement the transfer function associated with an ideal second-order differentiator (see inset). Again, the computer program can toggle between metasurface configurations such that second-order derivatives of signal envelopes are computed for different carriers (color-coded). **b**–**e**, Examples of four experimentally measured use cases corresponding to the four distinct indicated carrier frequencies. In each case, amplitude and phase of the system's transfer function are shown, as well as the experimentally measured output signal envelope upon injection of a Gaussian pulse. The displayed output signal envelopes are averaged over 20 acquisitions to alleviate the impact of measurement noise.

power incident signals without significant localized heating. Furthermore, the programmability of our setup regarding the CPA-EP frequency may be important for future absorption applications, too.

## Discussion

Besides speed and the ease of processing signals in their native domain, a compelling argument for wave-based computing is its energy efficiency. Indeed, a passive material can offer the desired transfer function without any energy consumption. However, a passive material cannot serve as the basis of a fair comparison with our technique since a passive material lacks all the features and advantages of reconfigurability that motivate our present work. Our device is quasi-passive: the programmable metasurface does not provide any energy to the wave and only requires minimal power, as low as a few μW per meta-atom[83], to maintain the meta-atom in the desired state or to flip it. Future metasurface technologies, such as those based on chalcogenide glasses, may even overcome the need for energy to maintain a state and only require minimal energy to flip the state. All this is in contrast to the proposed reconfigurable photonic interferometer-based

processor from Ref. [29] which, besides phase modulators, also intrinsically relies on a multitude of signal amplifiers.

To summarize, we demonstrated experimentally the essential steps to implement a meta-programmable over-the-air wave-based differentiator. The unprecedented fidelity and flexibility of our approach is enabled by not fabricating a carefully designed device with a single resonance and a few degrees of freedom but leveraging in situ the massive amount of degrees of freedom offered by a programmable metasurface inside a complex scattering system in order to tune scattering zeros onto the real frequency axis at will. The resulting in-situ adaptability overcomes the extreme sensitivity of analog differentiators to any sort of detuning which originates from the fact that they operate at a scattering anomaly associated with a diverging dwell time. We demonstrated with direct temporal measurements of the output signal that our technique can, first, toggle between operation at different carrier frequencies, second, parallelize the computational operation based on the wave equation's linearity, and, third, implement higher-order differentiation. Our approach can be extended to process spatially encoded information, to use over-moded random scattering systems based on guided waves, and to optical or acoustic scattering, as detailed in Supplementary

Note 10. Besides the implementation of analog differentiation, the unprecedented flexibility and precision of the implemented filters is relevant to applications in wideband defense systems such as cognitive radio.

Looking forward, we envision that a frequency sensing module is integrated with the entire system such that the latter can autonomously adapt itself to the incident signal; such a self-adaptive control of the metasurface configuration based on sensor measurements is easily implemented with current technology[100,101]. We also note that the switching time between metasurface configurations can be as low as 20 μs with state-of-the-art microcontrollers (see, for instance, Ref. [73]). Our proposed meta-programmable high-precision differentiator can furthermore serve as the pivotal ingredient of analog solvers of differential equations[102].

Another avenue that can prove fruitful is to explore potential benefits of continuously programmable meta-atoms[76] as well as multi-channel CPA[42,67] for meta-programmable analog differentiators. The former may drastically simplify gradient-based search methods for the optimal configuration, and hence be worthwhile the additional electronic burden. The latter may, according to indications in our ongoing work, facilitate imposing real-valued zeros in our system which potentially offsets the requirement for multi-channel amplitude and phase control, or an additional constraint to make the CPA eigenvector coincide with a predefined one[68].

## Methods

**Experimental setup**. A photographic image of the experimental setup from Fig. 1 is shown in Supplementary Fig. 2. A waveguide-to-coax adapter (RA13PBZ012-B-SMA-F) couples a guided wave to an electrically large disordered metallic box. Two programmable metasurfaces, each containing 76 meta-atoms, cover 16.2% of the cavity surface. Within a 400 MHz interval centered on 5.2 GHz, these metasurfaces can efficiently manipulate the scattered field. The metasurface operating band could be increased through refined metasurface designs or by combining multiple metasurfaces whose operating bands are centered on different frequencies. Each meta-atom offers independent 1-bit control over two orthogonal field polarizations, mimicking approximately Dirichlet or Neuman boundary conditions under normal incidence. Ohmic losses on the cavity walls result in a quality factor of the enclosure on the order of 410 such that approximately 21 modes overlap at a given frequency on average. Further details about the experimental setup and the characterization of the metasurface prototype are provided in Supplementary Note 5. Note that our concept is generic and could be implemented with any other programmable metasurface design, too.

**Determination of metasurface configurations**. Identifying a metasurface configuration that yields the desired scattering response, e.g., a real-valued scattering zero at a desired frequency, is an inverse problem. Conventional designs of analog differentiators are based on systems with a single or a few well-defined modes which can usually be described with reasonable accuracy through analytical models (e.g., using coupled-mode theory in Ref. [56]). In contrast, our technique relies on a purposefully perturbed random overmoded scattering system. As detailed in Supplementary Note 4, more than 20 modes overlap at any given frequency in our system, and given the irregular geometry of the metallic box, it is unfeasible to obtain the necessary information for each mode (frequency, width, and spatial pattern), let alone as a function of the programmable metasurface configuration, to formulate an analytical deterministic description.

For our proof-of-principle experiments, we solved this inverse problem via an iterative experimental trial-and-error algorithm detailed in Supplementary Note 5 which involved roughly 800 measurements with a vector network analyzer (Agilent Technologies PNA-L Network Analyzer N5230C, 0 dBm emitted power, 10 kHz intermediate-frequency bandwidth). In the future, the inverse design of metasurface configurations can be performed much faster and in software, once an artificial neural network has been trained to approximate the function that maps the configuration to the scattering response. References [79] and [103] already employ learned surrogate forward models to predict the parametrization of wireless channels through a programmable metasurface in quasi-free space and inside a complex scattering enclosure, respectively. Next-generation meta-atoms with fine-grained programmability (>1-bit) will expedite the optimization through their compatibility with continuous gradient-descent protocols. In certain settings without environmental perturbations during runtime, the identification of suitable metasurface configurations can be completed offline during a calibration phase and presents no burden during runtime. Generally speaking, when the optimization objectives are challenging and the available number of degrees of freedom is limited, in many applications one can find significant constraint relaxations such as not requiring the zero to occur at a specific frequency but within a certain interval.

**Time-domain experiments**. For a given test scenario involving a specific function $e(t)$ to be derived and a specific carrier $\omega_0$, we generated the waveform $e(t)e^{i\omega_0 t}$ with a signal generator (Aeroflex IFR 3416, 2 dBm emitted power, 33 MHz sampling rate) and toggled the metasurface to the configuration suitable for the chosen carrier. The input signal was injected via Port 1 of a circulator (PE83CR006; see Supplementary Fig. 4b), impinged on the system interface via Port 2 of the circulator, and the output signal which exited through Port 3 of the circulator was measured on an oscilloscope (SDA 816Zi-B 16 GHz Serial Data Analyzer, 40GS/s sampling rate). For the experiments on parallelized computing, a second signal generator (Agilent MXG Analog N5183A) generated the second input signal, and the two input signals were summed before being injected via Port 1 of the circulator. More details on the time-domain experiments can be found in Supplementary Note 5.

## Data availability

The data that support the findings of this study are available from the corresponding author upon reasonable request because all generated data is directly displayed in the figures without any additional processing. Access to the raw data can be gained by contacting the corresponding author via email (philipp.del-hougne@univ-rennes1.fr). We will do our best to respond to such requests within 14 days. Users of our data are requested to refer to this paper when publishing results partially or fully based on our data.

## Code availability

The codes used to optimize the metasurface configuration, as well as the codes used to generate the schematic illustrations in Fig. 1, Fig. 3a and Fig. 4a or the simple "GAUSSIAN", "POLYNOMIAL" and "SKYLINE" curves from Fig. 2c, f, i can be obtained by interested readers upon reasonable request by contacting the corresponding author via email (philipp.del-hougne@univ-rennes1.fr). We will do our best to respond to such requests within 14 days. Users of our codes are requested to refer to this paper when publishing results partially or fully based on our codes.

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

## Acknowledgements

The authors thank Steven M. Anlage, Thomas M. Antonsen, Matthieu Davy, Edward Ott and A. Douglas Stone for stimulating discussions, and François Yven for helpful exchanges regarding the time-domain measurement setup. J.S. and P.d.H. acknowledge funding from the European Union through the European Regional Development Fund (ERDF), and the French region of Brittany and Rennes Métropole through the CPER Project SOPHIE/STIC & Ondes. The metasurface prototypes were purchased from Greenerwave.

## Author contributions

P.d.H. conceived the project. J.S. and P.d.H. conducted the experiments. D.R.S. and P.d.H. interpreted the results. All authors contributed with thorough discussions. P.d.H. wrote the manuscript.

## Competing interests

The authors declare no competing interests.
