## [Peer Review File · Nature Communications]

Meta-Programmable Analog DifferentiatorREVIEWER COMMENTS

Reviewer #1 (Remarks to the Author):

The authors proposed a wave-based method for performing temporal differentiation of a microwave signal sent into a multi-mode disordered cavity, using a reconfigurable metasurface to bring a zero of the scattering matrix at the frequency of interest. Unlike previous works, the proposed technique leverages a programmable metasurface inside a 3D disordered metallic box for searching desired configurations that yields a desired scattering response. They also implemented higher-order differentiators and parallelize multiple differentiation processes.

Before I can be sure about my recommendation, I would like the authors to clarify or justify the following points:

1- What is needed is to match the input port at a single frequency. I failed to grasp why it's interesting to do this in a room, rather than with a tunable RLC circuit, for example.

2- This study demonstrated the operation of temporal differentiation. Is it also possible to differentiate in the space domain, as it is done in other works in the literature?

3- Due to the complexity of performing simulations of the system, identifying a specific metasurface configuration that yields the desired transfer function requires a tremendous number of measurements, especially for higher-order differentiation or other linear operations. This seems to be an important drawback of the proposed approach.

4- Realizing higher-order differentiation transfer functions, a second-order differentiator for example, directly with a single disordered cavity seems hard and needs more degrees of freedom (based on the authors' remark on page 15). For this reason, the authors proposed to cascade two first-order differentiator cavities using circulators. Firstly, realizing such transfer functions with other approaches such as passive or tunable metasurfaces is simple, refer to several proposals in this area. Secondly, cascading several distinct cavities to perform linear operations, second-order differentiation or simple integrodifferential equation seems somehow awkward in real scenarios: it's very bulky and needs some important equipment and accurate calibration.

5- The proposed technique to perform linear operations is active and needs some power to maintain the meta-atom in the desired state or to flip it. However, according to state-of-art in this area, performing linear operations with passive material-based systems needs no power.

Reviewer #2 (Remarks to the Author):

The authors propose temporal differentiators by random scattering system with tunable metamaterials in the microwave region. By iterative experimental trial-and-error algorithm to tuning the parameters, the V-shape transfer functions are realized at multiple carrier frequencies. Since the general temporal response for such random scattering systems has been proposed and investigated in Ref. 9, I generally find it hard to evaluate with confidence in its novelty giving only realizing the temporal differentiation. Below there are some of my concerns for the authors' consideration.

-If the system just achieves the differentiation at a single or simultaneously a few frequencies (Figs. 2 and 3), for such tunable metamaterial with much large degree of freedoms, what the advantages are compared with conventional microwave filters?

-The authors argue that the scheme to realize the differentiation can be applicable to other wave systems, for example optical one. However, in the other systems, the metamaterials are much more difficult to tune the scattering response than the microwave ones.

- It is not written clearly about what geometrical parameters of tunable metamaterials and how to design such metamaterials. What parameters are critical for the design?

Reviewer #1

The authors proposed a wave-based method for performing temporal differentiation of a microwave signal sent into a multi-mode disordered cavity, using a reconfigurable metasurface to bring a zero of the scattering matrix at the frequency of interest. Unlike previous works, the proposed technique leverages a programmable metasurface inside a 3D disordered metallic box for searching desired configurations that yields a desired scattering response. They also implemented higher-order differentiators and parallelize multiple differentiation processes.

OUR RESPONSE:

We would like to complement the reviewer's summary of our work and highlight that the key contribution of our work lies in the *programmability*: we demonstrate high-fidelity differentiation for any desired single or multiple carrier frequencies within a wide frequency range. To experimentally implement these unprecedented capabilities in terms of flexibility, we introduce a generic unconventional hardware implementation based on purposefully perturbed overmoded random scattering systems which we implement experimentally through a metasurface-programmable 3D disordered metallic box.

Before I can be sure about my recommendation, I would like the authors to clarify or justify the following points:

1- What is needed is to match the input port at a single frequency. I failed to grasp why it's interesting to do this in a room, rather than with a tunable RLC circuit, for example.

OUR RESPONSE:

We highly appreciate the referee's comment that turns out to be a common point of confusion about our approach (see also Reviewer #2's first comment). Our approach to tune a wave-chaotic system with many overlapping resonances as opposed to tuning a single/few resonance system is indeed unusual in the wave processing community. Our initial manuscript only briefly summarized the relative benefits of our approach

- high and statistically uniform density of zeros
- hundreds of degrees of freedom

to tune one or multiple zero(s) to the real frequency axis at the desired position(s) and it appears indeed very important to substantiate these points further.

To start, we would like to refine the referee's statement: what is needed is to *perfectly* match the input port at a single (or multiple) frequencies. In other words, the zero(s) of the transfer function must *lie exactly on the real frequency axis* at the desired real frequencies, rather than merely being "quite close" to the real frequency axis. This refinement of the statement may at first sight seem like a nuance because for many conventional notch filter applications, anything below -20dB or -30dB is fine. But in building a faithful differentiator, -30 dB is not good enough. We have added an entire Supplementary Note to illustrate the vulnerability of analog differentiators (Supplementary Note 2 in the revised version, the associated figure is

reproduced below). Therein, we clearly demonstrate that minute imperfections of the transfer function severely deteriorate the faithfulness of the differentiation operation, to the point that it is not satisfactory anymore.

Supplementary Figure 1. Vulnerability of analog differentiators. For different transfer functions (a-e), the top line shows the magnitude of the transfer function on a linear scale, the second line shows the magnitude of the transfer function on a logarithmic scale, the third line shows the phase of the transfer function, the fourth line shows the magnitude of the output signal's spectrum (for a Gaussian envelope as input signal), and the bottom line shows the envelope of the output signal (again for a Gaussian envelope as input signal). The different transfer functions correspond to: **a**, ideal; **b**, like ideal but with $A = 0.9975$; **c**, like ideal but with $A = 0.99$; **d**, like ideal but τ increased by a factor of 1.01; with $A = 0.99$; **e**, like ideal but amplitude of the transfer function for $\omega < \omega_0$ multiplied by a factor of 0.75.

Conventional tunable lumped-element filters (RLC circuits) do not have the same flexibility and precision as our setup and are therefore unable to *perfectly* place one or multiple zeros exactly at the desired real frequencies **in a programmable manner**. It is certainly possible to critically couple a simple RLC circuit at a fixed frequency to a transmission line, and by detuning one of its parameters, the zero of interest will certainly move around in the complex plane. But in moving around, the zero will drift away from the real-frequency axis. Only under special conditions of \mathcal{PT} -symmetry that are certainly not met by simple RLC circuits (the uncompensated presence of absorption already trivially breaks \mathcal{PT} -symmetry) one has a guarantee that the zeros are constrained to move along the real frequency axis (as long as they do not coalesce and form an exceptional point, see Refs. [1], [2]). Therefore, in a conventional tunable RLC circuit nothing guarantees that the zeros will not drift away from the

real frequency axis upon tuning. And a single or few degrees of freedom are insufficient to prevent this deterioration of the transfer function that is associated with a severe loss of fidelity of the differentiator – see the above figure and Supplementary Note 2. On paper, one may expect this tuning to be perfect if one does not account for non-idealities such as coupling dispersion, transmission-line length variation, parasitic coupling, and the properties of the microstrip lines. Incidentally, the same remark applies to the optical work from Ref. 29 of the main text. In any practical experiment, however, all these non-idealities are at play.

To substantiate this claim, we have thoroughly surveyed the literature on tunable lumped-element notch filters. The table below summarizes representative achieved transfer functions in the literature, and, upon simple visual inspection it is clearly obvious that none of these systems would yield a faithful programmable analog differentiator (see also our new Supplementary Note 2). It is important to note that this statement should not be taken as a criticism of these published works on tunable lumped-element notch filters: these devices were simply designed for different purposes where other metrics mattered and, for instance, oftentimes a dip of -20 dB or -30 dB was largely sufficient.

Reference	Achieved Transfer Functions	Comments
Ref. [3]		 • Deepest dip does not exceed -20 dB.
Ref. [4]		 • Dip depth depends on ω_0. • Deepest dip does not exceed -40 dB.
Ref. [5]		 • Dip depth depends on ω_0. • Deepest dip does not exceed -25 dB.
Ref. [6]		 • Dip depth depends on ω_0. • Deepest dip does not exceed -15 dB. • Dip shapes are distorted and deviate from those of a single zero close to the real frequency axis.

Ref. [7]		 • Dip depth depends on ω_0. • Deepest dip does not exceed -25 dB.
Ref. [8]		 • Dip depth depends on ω_0. • Deepest dip does not exceed -20 dB.

In such conventional few-resonance filters, the notch depth is often directly linked to the quality factor of the utilized resonators. Achieving very deep and sharp notch filters (as needed for differentiation but also applications including cognitive radio) is thus very difficult, and making the central frequency programmable even more so. In contrast, our approach implements these features with ease even though the composite quality factor of our chaotic cavity is not very high (~ 410 , see Supplementary Note 3).

For direct comparison, here are the transfer function from our manuscript (reproduced from Figure 2), showing that we achieve below -70 dB together with an abrupt π phase jump (see Supplementary Figure 1) for any frequency within a 0.35 GHz interval around 5.225 GHz:

To summarize, while we do not dispute that *perfectly* matching an input at a single frequency to an RLC circuit is feasible, the key contribution of our work is its flexibility and programmability: we can achieve the perfect zero (below -70 dB) of the transfer function for any (or multiple!) frequencies within a large interval. We are not aware of any tunable RLC circuit that offers these features. If the referee can point us toward any such work, we will happily look into relative advantages and disadvantages compared to our approach. Moreover, in light of the Reviewer's second comment, we note that a simple RLC circuit would not offer the possibility to perform spatial differentiator without designing additional antenna infrastructure.

The reviewer's question also led us to realize that we should point out that our generic concept could also be implemented in programmable overmoded random scattering system based on *guided* waves, such as a network of coupled transmission lines with complex connectivity (also known as graph). The programmability could then originate, for instance, from a series of phase shifters integrated into the connections between nodes of the graphs. While such a system would qualify for an implementation of our proposed concept, the required hardware seems inconvenient compared to that of our experiments based on a chaotic cavity with regard to any realistic application. The elegance of our experiments is that any metallic enclosure (toolbox, microwave oven, etc.) can be taken and only an ultrathin programmable metasurface needs to be attached to the walls. In the context of upcoming 6G wireless communication, many rooms which are also scattering enclosures at < 6 GHz will naturally be equipped with "reconfigurable intelligent surfaces" such that the entire hardware needed to implement our concept "over the air" is already there. In contrast, a guided approach would require one to carry around a network of transmission lines.

Finally, we would like to mention that the referee's very important remark not only helped us to substantiate a key point of our work that was previously not sufficiently clear; the remark has also brought to our attention that the unique capabilities of our technique (wide tunability in terms of number of notches and notch centers while guaranteeing ultraprofound notch depths) is also relevant to traditional tunable notch filter applications, notably in wideband defense systems, and we will explore such applications in future work.

OUR ACTIONS:

- We clarified in our manuscript's introduction why conventional tunable lumped-element filters are not capable of delivering the flexibility paired with "A clear route to address the aforementioned challenges are wave processors that can be reprogrammed *in situ*. A notable application thereof to temporal differentiation involved a photonic integrated interferometer equipped with programmable phase modulators[9]. On paper, it appears that a single degree of freedom is sufficient to introduce a relative phase shift of $(2m + 1)\pi$ at ω_0 between the two interferometer arms²⁹, yielding a perfect differentiator. In practice, however, the transfer function's magnitude minimum in Ref.²⁹ appears to be around 0.16; this system has a zero *close to* rather than *on* the real frequency axis, which jeopardizes a *linear* "V" shape and the associated *abrupt* π phase jump. Such limitations are inherent in the use of a single or few degrees of freedom in systems with a single or few resonances, and these limitations are also found in tunable microwave notch filters⁵⁴⁻⁵⁷.¹ The limitations originate from unaccounted non-idealities in real-life implementations such as coupling dispersion, transmission-

¹ Tunable microwave notch filters are developed for applications other than analog differentiation where different metrics matter, resulting in filter shapes that are typically not only insufficiently deep but also oftentimes deformed with respect to the shape needed for analog differentiation.

line length variation, parasitic coupling, and other properties of the microstrip lines. Using a single or few degrees of freedom, it is certainly possible to tune few-resonance systems such that the zeros move in the complex plane; however, only under special conditions of \mathcal{PT} -symmetry^{41,42} that are certainly not met by simple tunable notch filters (the uncompensated presence of absorption already trivially breaks \mathcal{PT} -symmetry) there is a guarantee that the zeros move exactly on the real frequency axis upon tuning. Hence, upon tuning simple notch filters in practice, zeros do not remain on the real frequency axis, and a few degrees of freedom tend to be insufficient to prevent that they drift away from the real frequency axis.

In contrast, our approach offers at least two orders of magnitude more degrees of freedom. This massive increase in programmability, together with the high density of zeros inherent to overmoded random scattering systems, makes it easy for us to perturb the system such that one of its zeros is placed, at a desired frequency, *on* the real frequency axis with extremely high precision (notch depth < -70 dB). Moreover, we can also switch to different functionalities and simultaneously create multiple zeros at arbitrary frequencies which is of importance for parallel wave processing at distinct frequencies, exploiting the wave equation's linearity⁵⁸. Unlike conventional electronic processors, a single wave processor can simultaneously process various streams of information encoded on independent (spectral, polarization, etc.) channels^{53,58-60}, directly multiplying its effective speed by the number of independent channels."

- In the above-cited new discussion, we include four new citations on representative tunable microwave notch filters.
- We added a new Supplementary Note 2 on the vulnerability of analog differentiators, including an additional Supplementary Figure that we reproduced above.
- We added Supplementary Note 5 about the experimentally measured notch depths.
- In the main text, we now call out more explicitly the vulnerability of analog differentiators and refer the reader to Supplementary Note 2:

"In this Article, instead of fabricating a carefully designed single-mode structure, we take an overmoded random scattering system as starting point and show that purposeful perturbations of its scattering properties, here with hundreds of degrees of freedom offered by an array of programmable meta-atoms^{3,4}, enable unprecedented flexibility and fidelity of wave-based differentiators by imposing that zeros of the scattering matrix lie on the real axis, at will and *in situ*."

"Consequently, minute imperfections severely deteriorate and rapidly undermine the faithfulness of an analog differentiator, to the point that it becomes unsatisfactory, as illustrated in detail in Supplementary Note 2."

"This massive increase in programmability, together with the high density of zeros inherent to overmoded random scattering systems, makes it easy for us to perturb the system such that one of its zeros is placed, at a desired frequency, *on* the real frequency axis with extremely high precision (notch depth < -70 dB). Moreover, we can also switch to different functionalities and simultaneously create multiple zeros at arbitrary frequencies ..."

- We refined the following statement in the main text:
"To that end, we must be able to perfectly place a zero on the real-frequency axis "on demand" at any desired frequency."
- The summary of our work in the discussion now points out more explicitly the difference of our approach from conventional RLC circuits:
"The unprecedented flexibility of our approach is enabled by not fabricating a carefully designed device with a single resonance and a few degrees of freedom but leveraging *in situ* the massive amount of degrees of freedom offered by a programmable metasurface

inside a complex scattering system in order to tune scattering zeros onto the real axis at will.“

- We added a comment to our discussion to point out the relevance of our setup as tunable filter to wideband defense systems:
“Besides the implementation of analog differentiation, the unprecedented flexibility and precision of the implemented filters is relevant to applications in wideband defense systems such as cognitive radio.“
- We added a new Supplementary Note 9 on generalizations of our concepts, which includes a subsection on programmable overmoded random scattering systems based on *guided waves* as discussed here.
- We added the attribute “over-the-air“ at three positions in the main text and call out the new Supplementary Note 9 in the discussion of our main text which elaborates on the possibility to transpose our concept to guided waves

2- This study demonstrated the operation of temporal differentiation. Is it also possible to differentiate in the space domain, as it is done in other works in the literature?

OUR RESPONSE:

Yes, our approach can also be applied to the differentiation of spatially encoded information, but that scenario is more relevant to optical signal processing than to the targeted microwave domain. We would like to thank the referee for this question which gives us an opportunity to evidence the generality of our proposed concept.

In our present work, we obtained the hallmark V-shaped spectrum associated with critical coupling by detuning the frequency f . However, the same V-shaped spectrum is obtained in the vicinity of the critical coupling condition by detuning any other parameter. For the case of spatial differentiation, we must detune the angle of incidence. Thus, if the input signal is encoded spatially over a range of angles of incidence, as opposed to temporally over a range of frequencies, then the setup from the new Supplementary Figure 9a (reproduced below) can perform spatial instead of temporal differentiation.

Specifically, if the signal whose derivative is to be computed is encoded spatially rather than temporally, then it will obliquely illuminate a perforated cavity wall as sketched in Supplementary Figure 9a below. The partial perforation of the cavity wall is easily implemented for different cavity types (see Supplementary Figures 9b,c) and such setups are routinely used for computational microwave imaging (see, e.g., Refs. [10], [11]). The scattering properties of the chaotic cavity are then tuned such that the incident wave critically couples to (i.e., is perfectly absorbed by) the cavity at a specific angle θ_0 and frequency f_0 , following the same optimization approach that we use in our manuscript.

To summarize, the only modification of our current setup that would be needed is to perforate a cavity wall such that signals can impinge over a continuous range of incidence angles. The input signal would thus not impinge through a mono-modal waveguide but as a wave propagating in free space. This is a trivial modification and the underlying physics (the

scattering anomaly with a zero of the scattering matrix on the real frequency axis) is exactly the same as for the temporally encoded signals we use.

Finally, we point out that such spatially encoded signals are very relevant to optical implementations of differentiators because in optical signals information is predominantly encoded in space (e.g., a camera image). In the microwave domain, however, signals of interest such as radar or WiFi encode the information in time, just as in our experiment. The reasons for these differences in predominant information encoding in different wave systems can be traced back to the associated hardware: in optics it is easy to manipulate or capture monochromatic wavefronts over large apertures with SLMs or CCDs, respectively, but controlling them in time is more demanding. In contrast, in the microwave domain it is relatively easy to modulate signals in time but constructing large coherent apertures is cumbersome because each antenna's RF chain is costly.

Supplementary Figure 9. Metaprogrammable *spatial* analog differentiator. a, Concept. b, Implementation in a 3D chaotic cavity from Ref. [10]. c, Implementation in a flat quasi-2D chaotic cavity from Ref. [12].

OUR ACTIONS:

- We added a comment to the discussion in our main text:

“Our approach can be extended to process spatially encoded information, to using overmoded random scattering systems based on *guided* waves, and to optical or acoustic scattering, as detailed in Supplementary Note 9.”
- We added a new Supplementary Note 9 on generalizations of our concepts, which includes a subsection on generalizations to spatially encoded information as discussed here.

3- Due to the complexity of performing simulations of the system, identifying a specific metasurface configuration that yields the desired transfer function requires a tremendous number of measurements, especially for higher-order differentiation or other linear operations. This seems to be an important drawback of the proposed approach.

OUR RESPONSE:

We agree with the referee that in our proof-of-concept experiment the identification of suitable metasurface configurations is somewhat cumbersome because we aim at demonstrating the physical feasibility; speeding up this one-off calibration step is outside our current manuscript's scope.

However, as mentioned in our Methods section, in the future we plan to learn forward models of the metasurface-programmable scattering parameters inside a chaotic cavity and leverage them as surrogate model for rapid inverse design of the metasurface configuration for a desired scattering functionality. The use of learned forward models is already well established in nanophotonics [13]–[15], and has also already been applied in the microwave domain to programmable metasurfaces in free space in Ref. [16]. We have ongoing work on implementing such rapid AI-assisted inverse design protocols for metasurface-programmable rich scattering environments.

The goal of our present work is to experimentally demonstrate the physical feasibility of our proposal. In certain settings without environmental perturbations during runtime, the identification of suitable metasurface configurations can be completed offline during a calibration phase and presents no burden during runtime. While the vast literature in nanophotonics evidences the great potential of AI-assisted inverse design for the fast identification of suitable metasurface configurations, this constitutes an entire research track in its own right and these algorithmic explorations are outside the scope and goals of the present work.

Another important point pointed out in our discussion is that if we use continuously as opposed to 1-bit programmable meta-atoms in the future, we can apply traditional gradient-descent techniques more efficiently which is currently hampered due to the 1-bit programmability constrain. This would also significantly reduce the number of measurements needed to identify a suitable configuration with our current iterative optimization approach.

Finally, we would like to point out once again that our technique is **not** merely an alternative to established methods that achieves similar results with a more cumbersome setup. Instead, our technique is the first experimental report of high-fidelity meta-programmable analog differentiation. Therefore, to date, the difficulties of our approach are unavoidable if the attractive features of programmability that we unlock for analog differentiators are to be available.

OUR ACTIONS:

- We added a further comment to the the Methods section in the main text.to clarify various aspects of the above discussion:

“Next-generation meta-atoms with fine-grained programmability (> 1-bit) will expedite the optimization through their compatibility with continuous gradient-descent protocols. In certain settings without environmental perturbations during runtime, the

identification of suitable metasurface configurations can be completed offline during a calibration phase and presents no burden during runtime.”

- We also added similar comments to the related Supplementary Note 4.

4- Realizing higher-order differentiation transfer functions, a second-order differentiator for example, directly with a single disordered cavity seems hard and needs more degrees of freedom (based on the authors’ remark on page 15). For this reason, the authors proposed to cascade two first-order differentiator cavities using circulators. Firstly, realizing such transfer functions with other approaches such as passive or tunable metasurfaces is simple, refer to several proposals in this area.

OUR RESPONSE:

While we agree with the referee that *passive (static)* metasurface implementations for second-order differentiation have been proposed, we are not aware of proposals or implementations with *programmable* operation frequency. If the referee can point us to specific references on the latter, in particular experimental reports, we will happily look into potential advantages and disadvantages thereof compared to our approach. At the current stage, our impression is that the unique flexibility of our approach, originating from the high uniform density of zeros combined with the high amount of degrees of freedom, is unique and that because simple metasurface approaches lack similar flexibility, they do not find it as easy to differentiate at any desired carrier (or multiple carriers).

Moreover, we would like to clarify that our arguments for why we did not manage to implement CPA-EPs are of speculative nature; there is a possibility that we simply did not identify the most suitable optimization metric but that our available hardware is in fact capable of implementing CPA-EPs.

OUR ACTIONS:

- No action possible/needed without explicit reference to published work on programmable analog higher-order differentiators.
- We formulated our hypothesis why we did not manage to tune our system to a CPA-EP more cautiously:
“However, preliminary tests following this route were unsuccessful, indicating that more degrees of freedom and/or more elaborate optimization protocols may be necessary.”

Secondly, cascading several distinct cavities to perform linear operations, second-order differentiation or simple integrodifferential equation seems somehow awkward in real scenarios: it's very bulky and needs some important equipment and accurate calibration.

OUR RESPONSE:

We respectfully disagree with the referee's comment regarding the integrability of our approach in realistic scenarios based on the bulkiness of the setup.

First, as discussed in a dedicated paragraph in our manuscript's introduction, our concept can equally well be implemented with a quasi-2D programmable chaotic cavity, yielding a flat and compact hardware implementation. The underlying hardware thereof is well-established and routinely used for computational imaging applications these days, see, for instance, Ref. [12] and Supplementary Figure 9c.

Second, and more importantly, however, we challenge the traditional view that integrability requires miniaturization. In the microwave domain, processing in the native signal domain (to avoid conversion to other analog or digital domains) results in bulky "miniaturized" devices that are still several centimeters or even decimeters large. In our manuscript, we introduce a new paradigm on integrability, arguing that a potentially bulky device that already exists for a different functionality (microwave oven to heat food, military toolbox to carry equipment, etc.) can be endowed with a second signal processing functionality. Thereby, implementing our approach only requires us to mount the ultrathin programmable metasurface anywhere on the walls of any metallic enclosure that exists for some other primary purpose. Then, we can judiciously program the metasurface and thereby the enclosure's scattering properties, in order to endow the enclosure with a second signal processing functionality. An important realistic scenario for the above is found within the context of 6G wireless networks which are expected to equip rooms with reconfigurable intelligent surfaces. In such settings, all the hardware that we need to implement our technique is already there.

Finally, we point out that there is no substantial additional difficulty in implementing higher-order differentiators with cascaded first-order differentiators compared to implementing a single first-order differentiator. A second-order differentiator only requires one to find two metallic enclosures (which by no means have to be identical or similar), and the addition of a circulator to cascade them. Generally speaking, the most commonly needed differentiator in practical applications is that of first order, sometimes also that of second order, but above second order is not a common need. Hence, imagining a cascade of many enclosures for a high-order differentiator is in general not a scenario in line with realistic needs. Note that in our manuscript we also suggest an alternative implementation of higher-order differentiators based on a single cavity using a simple delay loop that is suitable for input signals with sufficient temporal sparsity.

Similarly to our last remarks in response to the Reviewer's third comment, we would like to stress that our approach enables *unprecedented flexibility and programmability*. In scenarios without need for programmability, a simpler and more compact device can be used. But if programmable high-fidelity differentiation is desired, then there is no known alternative to our approach. Moreover, as elaborated in the preceding paragraphs, various methods can advance the integrability of our device, including an important paradigm shift that decouples integrability from miniaturization.

OUR ACTIONS:

- We added the following clarifying sentence to the paragraph in our introduction that discusses integrability:

“Thereby, we introduce a new perspective on integrability that decouples it from device volume and related miniaturization efforts.”

5- The proposed technique to perform linear operations is active and needs some power to maintain the meta-atom in the desired state or to flip it. However, according to state-of-art in this area, performing linear operations with passive material-based systems needs no power.

OUR RESPONSE:

We find the referee’s comparison of our work with non-programmable devices unfair because the latter are by definition static and lack all the features and advantages of programmability that motivate our study.

A fair comparison would benchmark our work against a system that implements *programmable* mathematical operations. As detailed in our introduction, this is only the case for Ref. 29 of the main text which relies not only on phase modulators (comparable to our reflection-programmable meta-atoms) but also intrinsically involves a multitude of signal amplifiers. In Ref. 29, the system thus by design delivers energy to the wave and can be considered “active”. This is very different in our work: the programmable meta-atoms do *not* provide any energy to the wave, they merely shape the scattering. For this reason, we cautiously used the term “quasi-passive” as opposed to “active” to describe our system in our manuscript’s discussion; indeed, the term “active” may carry connotations of adding energy to the waves which our system does not do.

Thus, our approach is advantageous in terms of energy consumption with respect to the state-of-the-art (which is Ref. 29 as opposed to non-programmable passive systems).

Moreover, we also argued that alternative designs of programmable metasurfaces based on chalcogenide glasses may even remove the need for the minimal amount of energy needed to maintain meta-atoms in their desired state, such that one would then only need minimal energy to flip the states of meta-atoms.

OUR ACTIONS:

- We now point out in the discussion that our system cannot be fairly compared to static passive systems:

“However, a passive material cannot serve as the basis of a fair comparison with our technique since a passive material lacks all the features and advantages of reconfigurability that motivate our present work.”

Reviewer #2

The authors propose temporal differentiators by random scattering system with tunable metamaterials in the microwave region. By iterative experimental trial-and-error algorithm to tuning the parameters, the V-shape transfer functions are realized at multiple carrier frequencies. Since the general temporal response for such random scattering systems has been proposed and investigated in Ref. 9, I generally find it hard to evaluate with confidence in its novelty giving only realizing the temporal differentiation.

OUR RESPONSE:

The referee evokes our previous work in Ref. 9 of the main text and expresses concern as to whether our present manuscript is significantly different. To clear out such concerns, we list below the key differences:

1. Vulnerability of implemented operation:

Ref. 9 implemented arbitrary vector-matrix multiplication as opposed to differentiation. Both are linear operations, but differentiation is an intrinsically much more vulnerable operation due to the time delay divergence (see discussion in our manuscript's introduction). Achieving a system with the transfer function of programmable differentiation (see also Supplementary Note 2 on the high level of "perfection" needed) and observing *in situ* (as opposed to simulating in software) that it performs the desired mathematical operation is hence a much greater experimental challenge that was not previously achieved.

2. Nature of input to mathematical operation

In Ref. 9, the input to the mathematical operation was encoded into the programmable metasurface configuration (and thus in a *discrete* and *spatial* manner) whereas in our present work the input is encoded into the time domain (and thus in a *continuous* and *spectral* manner). Note that Ref. 9 did not report anything related to "temporal response" because it presented a monochromatic technique without any time dependent fields.

3. Conversions between analog and digital

Our present manuscript implements an *all-analog* differentiation: The signal to be differentiated (from radar, wireless communication, ...) can be captured in its native domain by an antenna and be guided to directly impinge on our proposed system without any conversion between digital and analog, or various analog domains. This is a crucial feature of any analog processor: the speed and energy efficiency advantages are severely compromised once one starts to have to wait for energy-hungry conversions between digital and analog. Ref. 9 fundamentally relied on encoding the input for the mathematical operation digitally into the metasurface configuration. Even "worse", Ref. 9 relied on averaging the output over multiple realizations of the operation. There was hence a substantial amount of analog-digital conversions, associated with severe latency and energy consumption penalties, whereas our present work is all-analog. Thereby, our present work preserves the fundamental advantages of analog computing in terms of speed and energy efficiency.

The above-listed three features clearly distinguish our present work from Ref. 9. To summarize, the present work implements a much more vulnerable operation in the signals native domain, without any reliance on conversions between digital and analog domains.

OUR ACTIONS:

- We added Supplementary Note 2 to illustrate the vulnerability of analog differentiators.
- We edit the discussion of Ref. 9 in our introduction to highlight that Ref. 9 did not consider any time dependent fields:

“A related alternative view on the integrability of wave processors was introduced in Ref.⁹, but for less vulnerable spatially discrete **monochromatic** arbitrary matrix multiplications. Moreover, in Ref.⁹ the configuration of the programmable metasurface was interpreted as input, that is, the wave processor did not operate in the signal’s native domain, and furthermore averaging over multiple realizations was necessary as a consequence.”

Below there are some of my concerns for the authors’ consideration.

-If the system just achieves the differentiation at a single or simultaneously a few frequencies (Figs. 2 and 3), for such tunable metamaterial with much large degree of freedoms, what the advantages are compared with conventional microwave filters?

OUR RESPONSE:

We highly appreciate the referee’s comment that turns out to be a common point of confusion about our approach (see also Reviewer #1’s first comment). Our approach to tune a wave-chaotic system with many overlapping resonances as opposed to tuning a single/few resonance system is indeed unusual in the wave processing community. Our initial manuscript only briefly summarized the relative benefits of our approach

- high and statistically uniform density of zeros
- hundreds of degrees of freedom

to tune one or multiple zero(s) to the real frequency axis at the desired position(s) and it appears indeed very important to substantiate these points further.

To start, we would like to highlight the tremendous difficulty associated with an experimental implementation of a programmable analog differentiator: what is needed is to *perfectly* match the input port at a single (or multiple) frequencies. In other words, the zero(s) of the transfer function must *lie exactly on the real frequency axis* at the desired real frequencies, rather than merely being “quite close” to the real frequency axis. At first sight this may seem like a nuance because for many traditional notch filter applications, anything below -20dB or -30dB is fine. But in building a faithful differentiator, -30 dB is not good enough. We have added an entire supplementary note to illustrate the vulnerability of analog differentiators (Supplementary Note 2 in the revised version). Therein, we clearly demonstrate that minute imperfections of the transfer function severely deteriorate the faithfulness of the differentiation operation, to the point

that it is not satisfactory anymore. The new supplementary figure illustrating how minute imperfections severely degrade the differentiator performance is reproduced below.

Supplementary Figure 1. Vulnerability of analog differentiators. For different transfer functions (a-e), the top line shows the magnitude of the transfer function on a linear scale, the second line shows the magnitude of the transfer function on a logarithmic scale, the third line shows the phase of the transfer function, the fourth line shows the magnitude of the output signal's spectrum (for a Gaussian envelope as input signal), and the bottom line shows the envelope of the output signal (again for a Gaussian envelope as input signal). The different transfer functions correspond to: **a**, ideal; **b**, like ideal but with $A = 0.9975$; **c**, like ideal but with $A = 0.99$; **d**, like ideal but τ increased by a factor of 1.01; with $A = 0.99$; **e**, like ideal but amplitude of the transfer function for $\omega < \omega_0$ multiplied by a factor of 0.75.

Conventional microwave filters do not have the same flexibility and precision as our setup and are therefore unable to *perfectly* place one or multiple zeros exactly at the desired real frequencies in a *programmable* manner. It is probably possible to achieve critical coupling with high precision at a fixed frequency using a carefully crafted microwave filter, and by detuning one of its parameters, the zero of interest will certainly move around in the complex plane. But in moving around, the zero will drift away from the real-frequency axis. Only under special conditions of \mathcal{PT} -symmetry that are certainly not met by simple microwave filters (the uncompensated presence of absorption already trivially breaks \mathcal{PT} -symmetry) one has a guarantee that the zeros are constrained to move along the real frequency axis (as long as they do not coalesce and form an exceptional point, see Refs. [1], [2]). Therefore, in a

conventional tunable microwave filter nothing guarantees that the zeros will not drift away from the real frequency axis upon tuning. And a single or few degrees of freedom are insufficient to prevent this deterioration of the transfer function that is associated with a severe loss of fidelity of the differentiator – see the above figure and Supplementary Note 2.

To substantiate this claim, we have thoroughly surveyed the literature on tunable microwave notch filters. The table below summarizes the achieved transfer functions in the literature, and, upon simple visual inspection it is clearly obvious that none of these systems would yield a faithful programmable analog differentiator (see also our new Supplementary Note 2). It is important to note that this statement should not be taken as a criticism of these published works on tunable microwave notch filters: these devices were simply designed for different purposes where other metrics mattered and, for instance, oftentimes a dip of -20 dB or -30 dB was largely sufficient.

Reference	Achieved Transfer Functions	Comments
Ref. [17]		 • Dip depth depends on ω_0. • Deepest dip does not even reach -40 dB.
Ref. [18]		 • Very large tunability range. • Dip depth depends on ω_0. • Deepest dip does not even reach -35 dB.

Ref. [19]		 Dip depth does not exceed -25 dB.
Ref. [20]		 Dip depth depends on ω_0. Deepest dip does not exceed -20 dB.
Ref. [21]		 Dip depth does not even reach -20 dB. Dip shapes are distorted and deviate from those of a single zero close to the real frequency axis.
Ref. [22]		 Dip depth does mostly not exceed -40 dB. Dip shapes are distorted and deviate from those of a single zero close to the real frequency axis.

Ref. [23]		 • Very limited tunability of ω_0. • Relies on microwave-optics-microwave conversion. • Dip depth does mostly not exceed -40 dB.
Ref. [24]		 • Extraordinary tunability range. • Relies on microwave-optics-microwave conversion. • Dip depth does mostly not exceed -30 dB. • Dip shapes are asymmetric (see Supplementary Figure 1e regarding issues due to asymmetry).
Ref. [25]		 • Extraordinary dip depths. • Relies on manual mechanical tuning. • Dip shapes deviate from those of a single zero close to the real frequency axis.

For direct comparison, here are the transfer function from our manuscript (reproduced from Figure 2), showing that we achieve below -70 dB together with an abrupt π phase jump (see Supplementary Figure 1) for any frequency within a 0.35 GHz interval around 5.225 GHz:

The referee wonders why we use so many degrees of freedom to “just” impose one or multiple zeros at desired positions on the real frequency axis. In this respect, we note that the only theoretically proven result regarding how many degrees of freedom are needed is the following from Ref. [2]: one continuously tunable system parameter is enough to make a zero cross the real frequency axis. However, this is without any constraints, e.g., on the real frequency at which the zero crosses the horizontal axis. The theorem also does not make any claims about multiple zeros that simultaneously lie on the real frequency axis. Therefore, this theorem does not allow us to make any conclusions about how many degrees of freedom (continuously tunable or 1-bit) one needs to impose one or multiple zero(s) *at desired positions* on the real frequency axis. In the absence of any rigorous theory, we cannot judge whether the number of degrees of freedom that we use in our experiment is “too high” or not in sight of the constraints that we impose. But the fact that, to the best of our knowledge, no reports on tunable notch filters that would meet the high standards needed for programmable analog differentiation exist, suggests that using substantially fewer degrees of freedom is insufficient.

To summarize, the unique advantage of our setup is that we can perfectly place one or multiple zeros of the transfer function at any desired frequency, as evidenced by the fact that we can guarantee < -70 dB notch depth, an abrupt π phase jump and the typical linear-magnitude V shape for any desired frequency within a 0.35 GHz interval. Moreover, as evidenced in Figure 3 and Supplementary Figure 8, we have the additional flexibility to do this for more than one frequency at the same time. Such “perfection” is essential to implement a faithful programmable analog differentiator, as highlighted in the new Supplementary Note 2.

Finally, we would like to mention that the referee’s very important remark not only helped us to substantiate a key point of our work that was previously not sufficiently clear; the remark has also brought to our attention that the unique capabilities of our technique (wide tunability in terms of number of notches and notch centers while guaranteeing ultraprofound notch depths) may benefit certain traditional tunable notch filter applications, notably in the military domain, and we will explore such applications in future work.

OUR ACTIONS:

- We clarified in our manuscript’s introduction why conventional tunable lumped-element filters are not capable of delivering the flexibility paired with

“A clear route to address the aforementioned challenges are wave processors that can be reprogrammed *in situ*. A notable application thereof to temporal differentiation involved a photonic integrated interferometer equipped with programmable phase modulators[9]. On paper, it appears that a single degree of freedom is sufficient to introduce a relative phase shift of $(2m + 1)\pi$ at ω_0 between the two interferometer arms²⁹, yielding a perfect differentiator. In practice, however, the transfer function’s magnitude minimum in Ref.²⁹ appears to be around 0.16; this system has a zero *close to* rather than *on* the real frequency axis, which jeopardizes a *linear* “V” shape and the associated *abrupt* π phase jump. Such limitations are inherent in the use of a single or few degrees of freedom in systems with a single or few resonances, and these limitations are also found in tunable microwave notch filters^{54–57}.² The limitations originate from unaccounted non-idealities in real-life implementations such as coupling dispersion, transmission-line length variation, parasitic coupling, and other properties of the microstrip lines. Using a single or few degrees of freedom, it is certainly possible to tune few-resonance

² Tunable microwave notch filters are developed for applications other than analog differentiation where different metrics matter, resulting in filter shapes that are typically not only insufficiently deep but also oftentimes deformed with respect to the shape needed for analog differentiation.

systems such that the zeros move in the complex plane; however, only under special conditions of \mathcal{PT} -symmetry^{41,42} that are certainly not met by simple tunable notch filters (the uncompensated presence of absorption already trivially breaks \mathcal{PT} -symmetry) there is a guarantee that the zeros move exactly on the real frequency axis upon tuning. Hence, upon tuning simple notch filters in practice, zeros do not remain on the real frequency axis, and a few degrees of freedom tend to be insufficient to prevent that they drift away from the real frequency axis.

In contrast, our approach offers at least two orders of magnitude more degrees of freedom. This massive increase in programmability, together with the high density of zeros inherent to overmoded random scattering systems, makes it easy for us to perturb the system such that one of its zeros is placed, at a desired frequency, *on* the real frequency axis with extremely high precision (notch depth < -70 dB). Moreover, we can also switch to different functionalities and *simultaneously* create multiple zeros at *arbitrary* frequencies which is of importance for parallel wave processing at distinct frequencies, exploiting the wave equation's linearity⁵⁸. Unlike conventional electronic processors, a single wave processor can simultaneously process various streams of information encoded on independent (spectral, polarization, etc.) channels^{53,58-60}, directly multiplying its effective speed by the number of independent channels."

- In the above-cited new discussion, we include four new citations on representative tunable microwave notch filters.
- We added a new Supplementary Note 2 on the vulnerability of analog differentiators, including an additional Supplementary Figure that we reproduced above.
- We added Supplementary Note 5 about the experimentally measured notch depths.
- In the main text, we now call out more explicitly the vulnerability of analog differentiators and refer the reader to Supplementary Note 2:

"In this Article, instead of fabricating a carefully designed **single-mode** structure, we take an overmoded random scattering system as starting point and show that purposeful perturbations of its scattering properties, here with hundreds of degrees of freedom offered by an array of programmable meta-atoms^{3,4}, enable unprecedented flexibility and fidelity of wave-based differentiators by imposing that zeros of the scattering matrix lie on the real axis, at will and *in situ*."

"Consequently, minute imperfections severely deteriorate and rapidly undermine the faithfulness of an analog differentiator, to the point that it becomes unsatisfactory, as illustrated in detail in Supplementary Note 2."

"This massive increase in programmability, together with the high density of zeros inherent to overmoded random scattering systems, makes it easy for us to perturb the system such that one of its zeros is placed, at a desired frequency, *on* the real frequency axis with extremely high precision (notch depth < -70 dB). Moreover, we can also switch to different functionalities and *simultaneously* create multiple zeros at *arbitrary* frequencies ..."

- We refined the following statement in the main text:
"To that end, we must be able to **perfectly** place a zero on the real-frequency axis "on demand" at any desired frequency."
- The summary of our work in the discussion now points out more explicitly the difference of our approach from conventional RLC circuits:
"The unprecedented flexibility of our approach is enabled by not fabricating a carefully designed device with a single resonance and a few degrees of freedom but leveraging *in situ* the massive amount of degrees of freedom offered by a programmable metasurface inside a complex scattering system in order to tune scattering zeros onto the real axis at will."

- We added a comment to our discussion to point out the relevance of our setup as tunable filter to wideband defense systems:
 “Besides the implementation of analog differentiation, the unprecedented flexibility and precision of the implemented filters is relevant to applications in wideband defense systems such as cognitive radio.”
- We added a new Supplementary Note 9 on generalizations of our concepts, which includes a subsection on programmable overmoded random scattering systems based on *guided waves* as discussed here. This new Supplementary Note 9 is called out in the discussion of our main text.

-The authors argue that the scheme to realize the differentiation can be applicable to other wave systems, for example optical one. However, in the other systems, the metamaterials are much more difficult to tune the scattering response than the microwave ones.

OUR RESPONSE:

We appreciate the referee’s interest in our comment that our generic concept could also be implemented in optical, acoustic or elastic scattering.

First, we would like to highlight that our implementation in the microwave domain is **not** purely a proof-of-concept aimed ultimately at optical implementations. Indeed, the differentiation of information that is temporally encoded into microwave carriers is of direct practical relevance in wireless communication, radar, etc.

Second, while it is not easy to quantify how “much more difficult“ the tuning of scattering responses in optical, acoustic or elastic scattering is, we detail below a variety of approaches that other groups already found and experimentally implemented in optics and acoustics.

To start, we note that several well-established switchable diodes enable the implementation of tunable scattering responses akin to the microwave programmable metasurface we use at much higher frequencies in the THz and even infrared regimes. For example, Schottky diodes in the THz regime (see Ref. [26] and the website of Teratech Components Ltd: <http://www.teratechcomponents.com/>) and thermal VO₂ diodes in the infrared regime (see Refs. [27], [28]) can be used. Detailed design proposals for programmable meta-atoms based on these switchable diodes for operation around 350 GHz and 150 THz, respectively, can be found in the electronic supplementary material of Ref. [29].

Moreover, a wide range of very different mechanisms for tunable metamaterials have been reported in acoustic and optical scattering, some of which we list below:

Acoustics:

- magnetic-field controlled elastomers [30]
- tunable membranes [31]–[34]
- geometrically-tunable resonators [35]–[39]
- electrolysis-controlled microbubble arrays for ultrasound [40]

Optics:

- electro-optical modulation [41]
- phase-change materials [42]–[44]
- applied-voltage-sensitive graphene [45]–[47]
- all-optical modification of the spatial refractive index profile [48]
- acousto-optic modulation [49]
- spatial light modulators (if we consider complex medium + SLM as the scattering system) [50]
- mechanical actuation [51], [52] and computer-controlled mechanical perturbations [53]

Given that our experimental expertise lies mainly in the microwave domain, we refrain from judging whether and to what extent the above-listed techniques are more difficult than our microwave implementation. Nonetheless, the sheer existence of such a variety of diverse experimental reports on tunable scattering in acoustics and optics already suggests that our concept can be implemented with relative ease for optical or acoustic scattering, too.

OUR ACTIONS:

- We added a new Supplementary Note 9 on generalizations of our concepts, which includes a subsection on generalizations to optical and acoustic scattering as discussed here.

- It is not written clearly about what geometrical parameters of tunable metamaterials and how to design such metamaterials. What parameters are critical for the design?

OUR RESPONSE:

The referee inquires about critical design parameters of our tunable metamaterials. In fact, the elegance of our concept is that most specific parameters do not really matter. The exact design of our programmable metasurface prototype is provided in Supplementary Note 3 but we could have used any other prototype based on different design principles. What matters is that the programmable metasurface impacts as many rays as possible. In other words, if we have enough tuning knobs, we can always precisely move one or several of the densely and statistically uniformly spread zero(s) of the scattering matrix onto desired positions on the real frequency axis, irrespective of how exactly these tuning knobs function. Our original submission made this point only briefly in one sentence in Supplementary Note 3 and we thank the referee for pointing out that this point deserves more attention.

The contribution of our work lies in an unexpected deployment of a programmable metasurface inside a chaotic cavity, as opposed to lying in the metasurface design itself. However, for our specific application, if one was to design a customized metasurface, the following properties would be desirable:

- Meta-atom size:
Each meta-atom should have the largest possible scattering cross-section so that it impacts as many rays as possible. This usually leads to using roughly half-wavelength-

sized meta-atoms as in our prototype and is often counterintuitive for those who develop metasurfaces for wave control in free space (beamforming, etc.).

- Programmability:
The meta-atom programmability must be at least 1-bit, but multi-bit or continuous tuning is advantageous in terms of achieving more fine-grained control over the rays that interact with a given meta-atom. From the optimization algorithm's perspective, continuously tunable meta-atoms would allow us to implement efficient gradient-descent techniques. Of course, these advantages must be counterbalanced with the additional cost of implementing >1 bit programmability.
- Absorption:
The absorption of waves by the metasurface should ideally not be significantly stronger than by the metallic cavity walls.
- Number of meta-atoms:
The more meta-atoms we have, the more rays inside the cavity we can control and the easier we will find it to bring one (or several) zero(s) of the scattering matrix onto the real axis at the desired position(s).

But, to explicitly answer the referee's question, as long as we have enough meta-atoms, each individually programmable with at least 1-bit resolution, then our proposal can be implemented. There are thus no critical design parameters.

OUR ACTIONS:

- We added the following clarification to the main text:
"Our generic technique is not limited to implementations based on a specific programmable metasurface design, all that matters is that the programmable meta-atoms control as many rays as possible."
- We now explicitly state in the Methods section of the main text that the detailed design of the programmable metasurface is irrelevant for our technique:
"Note that our concept is generic and could be implemented with any other programmable metasurface design, too."
- We elaborate in Supplementary Note 3 on what desirable characteristics an ideal programmable metasurface has for our application.
"An ideal programmable metasurface for our purpose (i) interacts with as many rays as possible, meaning that each meta-atom has the largest possible scattering cross-section and the metasurface consists of as many meta-atoms as possible, (ii) the meta-atom programmability is as fine-grained as possible (but at least 1-bit), and (iii) insertion of the metasurface into the chaotic enclosure does not significantly alter the amount of absorption."

References

- [1] A.-S. B.-B. Dhia, L. Chesnel, and V. Pagneux, "Trapped modes and reflectionless modes as eigenfunctions of the same spectral problem," *Proc. R. Soc. A.*, vol. 474, no. 2213, p. 20180050, May 2018, doi: 10.1098/rspa.2018.0050.
- [2] W. R. Sweeney, C. W. Hsu, and A. D. Stone, "Theory of reflectionless scattering modes," *Phys. Rev. A*, vol. 102, no. 6, p. 063511, Dec. 2020, doi: 10.1103/PhysRevA.102.063511.
- [3] Y.-C. Ou and G. M. Rebeiz, "Lumped-Element Fully Tunable Bandstop Filters for Cognitive Radio Applications," *IEEE Trans. Microwave Theory Techn.*, vol. 59, no. 10, pp. 2461–2468, Oct. 2011, doi: 10.1109/TMTT.2011.2160965.
- [4] J.-C. S. Chieh and J. Rowland, "A fully tunable C-band reflectionless bandstop filter using L-resonators," in *Proc. EuMC*, London, United Kingdom, Oct. 2016, pp. 131–133. doi: 10.1109/EuMC.2016.7824295.
- [5] A. Anand, Y. Liu, and X. Liu, "Substrate-integrated octave-tunable combline bandstop filter with surface mount varactors," in *Proc. IWS*, X'ian, China, Mar. 2014, pp. 1–4. doi: 10.1109/IEEE-IWS.2014.6864203.
- [6] C.-H. Ko and G. M. Rebeiz, "A 1.4–2.3-GHz Tunable Diplexer Based on Reconfigurable Matching Networks," *IEEE Trans. Microwave Theory Techn.*, vol. 63, no. 5, pp. 1595–1602, May 2015, doi: 10.1109/TMTT.2015.2411605.
- [7] S.-W. Jeong and J. Lee, "Frequency- and Bandwidth-Tunable Bandstop Filter Containing Variable Coupling Between Transmission Line and Resonator," *IEEE Trans. Microwave Theory Techn.*, vol. 66, no. 2, pp. 943–953, Feb. 2018, doi: 10.1109/TMTT.2017.2756963.
- [8] P.-L. Benson, "Tunable lumped element notch filter for UHF communications systems," Stellenbosch University, 2018.
- [9] W. Liu *et al.*, "A fully reconfigurable photonic integrated signal processor," *Nat. Photonics*, vol. 10, no. 3, pp. 190–195, Mar. 2016, doi: 10.1038/nphoton.2015.281.
- [10] T. Sleasman, M. F. Imani, J. N. Gollub, and D. R. Smith, "Microwave Imaging Using a Disordered Cavity with a Dynamically Tunable Impedance Surface," *Phys. Rev. Applied*, vol. 6, no. 5, Art. no. 5, Nov. 2016, doi: 10.1103/PhysRevApplied.6.054019.
- [11] M. F. Imani, T. Sleasman, and D. R. Smith, "Two-Dimensional Dynamic Metasurface Apertures for Computational Microwave Imaging," *IEEE Antennas Wirel. Propag. Lett.*, vol. 17, no. 12, Art. no. 12, Dec. 2018, doi: 10.1109/LAWP.2018.2873131.
- [12] T. Sleasman, M. F. Imani, A. V. Diebold, M. Boyarsky, K. P. Trofatter, and D. R. Smith, "Implementation and characterization of a two-dimensional printed circuit dynamic metasurface aperture for computational microwave imaging," *IEEE Trans. Antennas Propag.*, vol. 69, no. 4, p. 2151, Oct. 2020, doi: 10.1109/TAP.2020.3027188.
- [13] J. Peurifoy *et al.*, "Nanophotonic particle simulation and inverse design using artificial neural networks," *Sci. Adv.*, p. 8, Jun. 2018, doi: 10.1126/sciadv.aar4206.
- [14] P. R. Wiecha, A. Arbouet, C. Girard, and O. L. Muskens, "Deep learning in nano-photonics: inverse design and beyond," *Photonics Res.*, vol. 9, no. 5, p. B182, May 2021, doi: 10.1364/PRJ.415960.
- [15] C. C. Nadell, B. Huang, J. M. Malof, and W. J. Padilla, "Deep learning for accelerated all-dielectric metasurface design," *Opt. Express*, vol. 27, no. 20, Art. no. 20, Sep. 2019, doi: 10.1364/OE.27.027523.
- [16] H.-Y. Li *et al.*, "Intelligent Electromagnetic Sensing with Learnable Data Acquisition and Processing," *Patterns*, vol. 1, no. 1, Art. no. 1, Apr. 2020, doi: 10.1016/j.patter.2020.100006.
- [17] I. Gil, J. García-García, J. Bonache, F. Martín, M. Sorolla, and R. Marqués, "Varactor-loaded split ring resonators for tunable notch filters at microwave frequencies," *Electron. Lett.*, vol. 40, no. 21, p. 1347, Oct. 2004, doi: 10.1049/el:20046389.
- [18] Z. Wu, Y. Shim, and M. Rais-Zadeh, "Miniaturized UWB Filters Integrated With Tunable Notch Filters Using a Silicon-Based Integrated Passive Device Technology," *IEEE Trans. Microwave Theory Techn.*, vol. 60, no. 3, pp. 518–527, Mar. 2012, doi: 10.1109/TMTT.2011.2178428.

- [19] V. Urruchi, C. Marcos, J. Torrecilla, J. M. Sánchez-Pena, and K. Garbat, "Note: Tunable notch filter based on liquid crystal technology for microwave applications," *Rev. Sci. Instrum.*, vol. 84, no. 2, p. 026102, Feb. 2013, doi: 10.1063/1.4790555.
- [20] J. Torrecilla *et al.*, "Microwave Tunable Notch Filter Based on Liquid Crystal Using Spiral Spurline Technology," *Microw. Opt. Technol. Lett.*, vol. 55, no. 10, pp. 2420–2423, Oct. 2013, doi: 10.1002/mop.27812.
- [21] L. Kurra, M. P. Abegaonkar, A. Basu, and S. K. Koul, "Switchable and Tunable Notch in Ultra-Wideband Filter Using Electromagnetic Bandgap Structure," *IEEE Microw. Wireless Compon. Lett.*, vol. 24, no. 12, pp. 839–841, Dec. 2014, doi: 10.1109/LMWC.2014.2363020.
- [22] C.-W. Tang and W.-C. Chen, "A Compact Tunable Notch Filter With Wide Constant Absolute Bandwidth," *IEEE Microw. Wireless Compon. Lett.*, vol. 25, no. 3, pp. 151–153, Mar. 2015, doi: 10.1109/LMWC.2014.2382671.
- [23] Weiqi Xue, S. Sales, J. Mork, and J. Capmany, "Widely Tunable Microwave Photonic Notch Filter Based on Slow and Fast Light Effects," *IEEE Photon. Technol. Lett.*, vol. 21, no. 3, pp. 167–169, Feb. 2009, doi: 10.1109/LPT.2008.2009468.
- [24] M. S. Rasras *et al.*, "Demonstration of a Tunable Microwave-Photonic Notch Filter Using Low-Loss Silicon Ring Resonators," *J. Lightwave Technol.*, vol. 27, no. 12, pp. 2105–2110, Jun. 2009, doi: 10.1109/JLT.2008.2007748.
- [25] Y. Y. Danilov, G. G. Denisov, M. A. Khozin, A. Panin, and Y. Rodin, "Millimeter-Wave Tunable Notch Filter Based on Waveguide Extension for Plasma Diagnostics," *IEEE Trans. Plasma Sci.*, vol. 42, no. 6, pp. 1685–1689, Jun. 2014, doi: 10.1109/TPS.2014.2318352.
- [26] W. C. B. Peatman, P. A. D. Wood, D. Porterfield, T. W. Crowe, and M. J. Rooks, "Quarter-micrometer GaAs Schottky barrier diode with high video responsivity at 118 μm ," *Appl. Phys. Lett.*, vol. 61, no. 3, pp. 294–296, Jul. 1992, doi: 10.1063/1.107944.
- [27] A. S. Barker, H. W. Verleur, and H. J. Guggenheim, "Infrared Optical Properties of Vanadium Dioxide Above and Below the Transition Temperature," *Phys. Rev. Lett.*, vol. 17, no. 26, pp. 1286–1289, Dec. 1966, doi: 10.1103/PhysRevLett.17.1286.
- [28] A. Ghanekar, J. Ji, and Y. Zheng, "High-rectification near-field thermal diode using phase change periodic nanostructure," *Appl. Phys. Lett.*, vol. 109, no. 12, p. 123106, Sep. 2016, doi: 10.1063/1.4963317.
- [29] L. Li *et al.*, "Electromagnetic reprogrammable coding-metasurface holograms," *Nat. Commun.*, vol. 8, no. 1, Art. no. 1, Dec. 2017, doi: 10.1038/s41467-017-00164-9.
- [30] X. Chen, X. Xu, S. Ai, H. Chen, Y. Pei, and X. Zhou, "Active acoustic metamaterials with tunable effective mass density by gradient magnetic fields," *Appl. Phys. Lett.*, vol. 105, no. 7, p. 071913, Aug. 2014, doi: 10.1063/1.4893921.
- [31] S. Xiao, G. Ma, Y. Li, Z. Yang, and P. Sheng, "Active control of membrane-type acoustic metamaterial by electric field," *Appl. Phys. Lett.*, vol. 106, no. 9, p. 091904, Mar. 2015, doi: 10.1063/1.4913999.
- [32] Z. Chen *et al.*, "A tunable acoustic metamaterial with double-negativity driven by electromagnets," *Sci. Rep.*, vol. 6, no. 1, p. 30254, Jul. 2016, doi: 10.1038/srep30254.
- [33] G. Ma, X. Fan, P. Sheng, and M. Fink, "Shaping reverberating sound fields with an actively tunable metasurface," *Proc. Natl. Acad. Sci. USA*, vol. 115, no. 26, pp. 6638–6643, Jun. 2018, doi: 10.1073/pnas.1801175115.
- [34] W. Ao, J. Ding, L. Fan, and S. Zhang, "A robust actively-tunable perfect sound absorber," *Appl. Phys. Lett.*, vol. 115, no. 19, p. 193506, Nov. 2019, doi: 10.1063/1.5123423.
- [35] Z. Tian *et al.*, "Programmable Acoustic Metasurfaces," *Adv. Funct. Mater.*, vol. 29, no. 13, p. 1808489, Mar. 2019, doi: 10.1002/adfm.201808489.
- [36] W. K. Cao *et al.*, "Tunable Acoustic Metasurface for Three-Dimensional Wave Manipulations," *Phys. Rev. Applied*, vol. 15, no. 2, p. 024026, Feb. 2021, doi: 10.1103/PhysRevApplied.15.024026.
- [37] C. Zhang *et al.*, "A reconfigurable active acoustic metalens," *Appl. Phys. Lett.*, vol. 118, no. 13, p. 133502, Mar. 2021, doi: 10.1063/5.0045024.

- [38] S.-D. Zhao, A.-L. Chen, Y.-S. Wang, and C. Zhang, "Continuously Tunable Acoustic Metasurface for Transmitted Wavefront Modulation," *Phys. Rev. Applied*, vol. 10, no. 5, p. 054066, Nov. 2018, doi: 10.1103/PhysRevApplied.10.054066.
- [39] S.-W. Fan, S.-D. Zhao, A.-L. Chen, Y.-F. Wang, B. Assouar, and Y.-S. Wang, "Tunable Broadband Reflective Acoustic Metasurface," *Phys. Rev. Applied*, vol. 11, no. 4, p. 044038, Apr. 2019, doi: 10.1103/PhysRevApplied.11.044038.
- [40] Z. Ma *et al.*, "Spatial ultrasound modulation by digitally controlling microbubble arrays," *Nat. Commun.*, vol. 11, no. 1, p. 4537, Dec. 2020, doi: 10.1038/s41467-020-18347-2.
- [41] A. M. Stolyarov, L. Wei, F. Sorin, G. Lestoquoy, J. D. Joannopoulos, and Y. Fink, "Fabrication and characterization of fibers with built-in liquid crystal channels and electrodes for transverse incident-light modulation," *Appl. Phys. Lett.*, vol. 101, no. 1, Art. no. 1, Jul. 2012, doi: 10.1063/1.4733319.
- [42] B. Gholipour, J. Zhang, K. F. MacDonald, D. W. Hewak, and N. I. Zheludev, "An All-Optical, Non-volatile, Bidirectional, Phase-Change Meta-Switch," *Adv. Mater.*, vol. 25, no. 22, pp. 3050–3054, Jun. 2013, doi: 10.1002/adma.201300588.
- [43] D. Wang *et al.*, "Switchable Ultrathin Quarter-wave Plate in Terahertz Using Active Phase-change Metasurface," *Sci. Rep.*, vol. 5, no. 1, p. 15020, Dec. 2015, doi: 10.1038/srep15020.
- [44] Q. Wang *et al.*, "Optically reconfigurable metasurfaces and photonic devices based on phase change materials," *Nat. Photonics*, vol. 10, no. 1, Art. no. 1, Jan. 2016, doi: 10.1038/nphoton.2015.247.
- [45] L. Ju *et al.*, "Graphene plasmonics for tunable terahertz metamaterials," *Nat. Nanotechnol.*, vol. 6, no. 10, pp. 630–634, Oct. 2011, doi: 10.1038/nnano.2011.146.
- [46] Y. Yao *et al.*, "Broad Electrical Tuning of Graphene-Loaded Plasmonic Antennas," *Nano Lett.*, vol. 13, no. 3, pp. 1257–1264, Mar. 2013, doi: 10.1021/nl3047943.
- [47] Y.-W. Huang *et al.*, "Gate-Tunable Conducting Oxide Metasurfaces," *Nano Lett.*, vol. 16, no. 9, pp. 5319–5325, Sep. 2016, doi: 10.1021/acs.nanolett.6b00555.
- [48] R. Bruck *et al.*, "All-optical spatial light modulator for reconfigurable silicon photonic circuits," *Optica*, vol. 3, no. 4, Art. no. 4, Apr. 2016, doi: 10.1364/OPTICA.3.000396.
- [49] M. Bello-Jiménez, E. Hernández-Escobar, A. Camarillo-Avilés, O. Pottiez, A. Díez, and M. V. Andrés, "Actively mode-locked all-fiber laser by 5 MHz transmittance modulation of an acousto-optic tunable bandpass filter," *Laser Phys. Lett.*, vol. 15, no. 8, Art. no. 8, Aug. 2018, doi: 10.1088/1612-202X/aac9d8.
- [50] M. W. Matthès, P. del Hougne, J. de Rosny, G. Lerosey, and S. M. Popoff, "Optical complex media as universal reconfigurable linear operators," *Optica*, vol. 6, no. 4, Art. no. 4, Apr. 2019, doi: 10.1364/OPTICA.6.000465.
- [51] J. Y. Ou, E. Plum, L. Jiang, and N. I. Zheludev, "Reconfigurable Photonic Metamaterials," *Nano Lett.*, vol. 11, no. 5, pp. 2142–2144, May 2011, doi: 10.1021/nl200791r.
- [52] J.-Y. Ou, E. Plum, J. Zhang, and N. I. Zheludev, "An electromechanically reconfigurable plasmonic metamaterial operating in the near-infrared," *Nat. Nanotechnol.*, vol. 8, no. 4, Art. no. 4, Apr. 2013, doi: 10.1038/nnano.2013.25.
- [53] S. Resisi, Y. Viernik, S. M. Popoff, and Y. Bromberg, "Wavefront shaping in multimode fibers by transmission matrix engineering," *APL Photonics*, vol. 5, no. 3, Art. no. 3, Mar. 2020, doi: 10.1063/1.5136334.

REVIEWER COMMENTS

Reviewer #1 (Remarks to the Author):

The authors have addressed my comments satisfactorily.

Reviewer #2 (Remarks to the Author):

Thank the authors for their responses and for resolving some of the issues brought up in previous comments. Indeed, it has shown the advantages that the highly deep V-shape transfer functions can be simultaneously realized at multiple carrier frequencies by iterative experimental trial-and-error algorithm to tuning the parameters, in random scattering systems. However, about to comment 3, it not clearly shows the physics of the filter. Otherwise, these results could be strongly or even accidentally dependent on the numerical optimization. It is better to compare the physics to multiple mode coupling spatial differentiator [e.g. Optics letters 43, 5893, 2018], which also realizes the differentiation in multiple frequencies. In addition, is it truly no up or down frequency limitation, as states that "create multiple zeros at arbitrary frequencies"?

Reviewer #1

The authors have addressed my comments satisfactorily.

OUR RESPONSE:

We are delighted to read that our previous rebuttal letter fully resolved all concerns and answered all questions previously raised by the reviewer.

Reviewer #2

Thank the authors for their responses and for resolving some of the issues brought up in previous comments. Indeed, it has shown the advantages that the highly deep V-shape transfer functions can be simultaneously realized at multiple carrier frequencies by iterative experimental trial-and-error algorithm to tuning the parameters, in random scattering systems.

OUR RESPONSE:

We are delighted to read that our previous rebuttal letter helped to resolve the reviewer's concerns and questions.

However, about to comment 3, it not clearly shows the physics of the filter. Otherwise, these results could be strongly or even accidentally dependent on the numerical optimization. It is better to compare the physics to multiple mode coupling spatial differentiator [e.g. Optics letters 43, 5893, 2018], which also realizes the differentiation in multiple frequencies.

OUR RESPONSE:

The reviewer recommends an analysis of the physics underlying our technique based on a coupled-mode theory and indicates a reference in which such an analysis was performed.

To start, we thank the reviewer for bringing to our attention this very interesting paper. In the cited paper, the authors numerically explore a metasurface that simultaneously differentiates various signals carried by different wavelengths, and that outputs each through a different channel. Compared to this work, here are some key advantages of our work:

1. Fidelity. The numerical results in the cited paper show good differentiation performance but clear deviations from the ideal differentiator output are apparent (see Fig. 3 in the cited paper). If the results from the cited paper were to be realized experimentally, the performance would certainly be deteriorated due to fabrication inaccuracies and environmental perturbations. Our *experimental* results achieve a significantly higher fidelity than the numerical results in the cited paper (e.g., abruptness of the phase

jump; for the Gaussian input pulse considered in the cited paper: symmetry of output with respect to critically coupled wavelength, and zero signal at the latter). The reason is, of course, related to the unprecedented flexibility offered by our setup (more on this below).

2. **Programmability.** A crucial contribution of our work is to enable *programmable* analog differentiation with high fidelity. This is tremendously difficult if one operates with system designs based on few resonances (as in the cited paper) because upon tuning the zeros drift away from the real frequency axis (see our previous rebuttal letter). While the cited paper offers no programmability, our paper does.
3. **Experiments.** Experimental validation adds significant value. In our work, we not only measure the transfer function of our experimental system, we go further and inject in situ different input signals and record the system's output signals, in order to offer direct evidence of its differentiation performance.

It is very important to note the **fundamental difference in the physics** behind conventional wave-based differentiators (including the cited paper) and our technique.

- **Conventional techniques** carefully design a system with a single or a few well-defined resonance(s)/mode(s) in order to implement the desired critical coupling condition. Such "simple" scattering systems can be described with reasonable accuracy through analytical models. A common example is the coupled-mode theory, and the cited paper used this approach to describe and optimize the system design.
- **Our technique** relies on a purposefully perturbed random overmoded scattering system. Random scattering systems like ours display wave-chaotic behavior (see, e.g., Stöckmann, H. J. (2000). *Quantum Chaos: An Introduction*). Moreover, as detailed in our Supplementary Note 3, more than 20 modes overlap at any given frequency in our irregularly shaped cavity, each mode having a speckle-like spatial profile. Obtaining the necessary information to deterministically describe so many coupled modes (resonance frequency, resonance width, spatial mode pattern) with reasonable accuracy is unfeasible. Instead, in the wave chaos community, it is customary to describe such systems statistically. Indeed, a great deal of results about the statistics of many random realizations of a wave-chaotic cavity like ours are known. But we *purposefully perturb* our wave-chaotic system using the programmable metasurface, because we seek a rare event of the chaotic system's statistical distribution that corresponds to "critical coupling"¹. For all these reasons, neither a deterministic analytical nor a statistical approach is available to describe our system. Therefore, we optimize our system in situ with a trial and error approach.

This difference explains the origin of the unprecedented performance of our system in terms of (i) fidelity and (ii) programmability. In our case, the complex plane is densely populated with zeros, and we have hundreds of degrees of freedom at our disposal to move one or multiple of them to desired positions exactly on the real frequency axis. Conventional approaches have one or a few zero(s) in the complex plane, and use a very limited number of degrees of freedom to control the location of the zero(s) in the complex plane.

Our work involves no numerical optimization. As in any inverse design process without guarantee for finding the global optimum, running the optimization twice certainly does not yield exactly the same result. This is well known in all areas of science involving inverse design. But, as discussed in our Supplementary Note 4, we observe that running the optimization process again yields a different local optimum of a similar quality.

¹ The concept of critical coupling usually refers to systems with a single isolated resonance and its generalization to systems with overlapping resonances is non-trivial.

The physics of perfect absorption in random overmoded scattering systems is well established in the literature. A comprehensive description of the underlying theory can be found, for instance, in Ref.⁴² of our manuscript. In Section II.E of this reference, Stone and coworkers develop a coupled-mode analysis that paints a clear picture of the physics at play when the perfect absorption condition is satisfied in the random overlapping regime. We kindly refer the reviewer to this reference for details on the physical picture. In short, the simple picture of critical coupling from the isolated resonance regime breaks down substantially but nonetheless a rigorous theory of reflectionless scattering modes in random overmoded scattering systems exists. However, the theoretical developments from Ref.⁴² cannot directly serve as design tool in our experiment because, as stated above, the necessary information about the 20+ overlapping modes (frequency, width, spatial pattern + their dependence on the programmable metasurface configuration) is not available.

To summarize, a deterministic analytical description of our scattering system is not feasible due to its randomness and overmodedness. The reviewer's remark helped us to note that these reasons need further clarification for the benefit of the broad readership.

OUR ACTION:

- We added the following explanation to the subsection "Determination of metasurface configurations" in our Methods section:
"Conventional designs of analog differentiators are based on systems with a single or a few well-defined modes which can usually be described with reasonable accuracy through analytical models (e.g., using coupled-mode theory in Ref.⁵⁸). In contrast, our technique relies on a purposefully perturbed random overmoded scattering system. As detailed in Supplementary Note 3, more than 20 modes overlap at any given frequency in our system, and given the irregular geometry of the metallic box, it is unfeasible to obtain the necessary information for each mode (frequency, width, and spatial pattern), let alone as function of the programmable metasurface configuration, to formulate an analytical deterministic description."
- We now cite the reference suggested by the reviewer (Ref. 58 in our revised manuscript) whenever we cite references on parallelized wave processing as well as in the above-cited paragraph in the context of coupled-mode theory.

In addition, is it truly no up or down frequency limitation, as states that "create multiple zeros at arbitrary frequencies"?

OUR RESPONSE:

We appreciate the reviewer's interest in the programmability of our technique. The programmability relies on our ability to tune the wave-chaotic cavity's scattering properties with the hundreds of degrees of freedom offered by our programmable metasurface. The latter has a wide but obviously finite range over which it can modulate the electromagnetic field. We kindly refer the reviewer to our Supplementary Note 3 where in particular Supplementary Figure 3 offers a thorough characterization of our programmable metasurface. Certainly for any frequency within a 0.4 GHz interval centered on 5.2 GHz we can impose a zero with very high fidelity.

If our goal was to operate over an interval much larger than the operating band of the programmable metasurface, we could simply use multiple metasurfaces centered on different operating frequencies.

To summarize, we thank the reviewer for alerting us to the need to clarify in the cited statement that the arbitrary frequency should of course fall into the programmable metasurface's operating range.

OUR ACTION:

- We added a clarifying comment to the cited sentence:
“Moreover, we can also switch to different functionalities and *simultaneously* create multiple zeros at *arbitrary* frequencies (within the programmable metasurface's operating band) which is of importance for parallel wave processing at distinct frequencies, exploiting the wave equation's linearity^{58,59}.”
- We added the following clarification to the subsection “Experimental setup” of our Methods section:
“Within a 400 MHz interval centered on 5.2 GHz, these metasurfaces can efficiently manipulate the scattered field. The metasurface operating band could be increased through refined metasurface designs or by combining multiple metasurfaces whose operating bands are centered on different frequencies.”

REVIEWERS' COMMENTS

Reviewer #2 (Remarks to the Author):

The authors have answered all the questions in details and I would like to recommend the manuscript publishing as present.

Reviewer #2

The authors have answered all the questions in details and I would like to recommend the manuscript publishing as present.

OUR RESPONSE:

We are delighted to read that our previous rebuttal letter fully resolved all concerns and answered all questions previously raised by the reviewer.